# The *Plasmodium* NOT1-G paralogue is an essential regulator of sexual stage maturation and parasite transmission

Kevin J. Hart[1], B. Joanne Power[1]⊙, Kelly T. Rios[1]⊙, Aswathy Sebastian[2], Scott E. Lindner[1] *

1 Department of Biochemistry and Molecular Biology, The Huck Center for Malaria Research, Pennsylvania State University, University Park, Pennsylvania, United States of America, 2 Huck Institutes of the Life Sciences, Pennsylvania State University, University Park, Pennsylvania, United States of America

⊙ These authors contributed equally to this work.
* Scott.Lindner@psu.edu

**Data Availability Statement:** RNA-seq data reported here is available through the GEO depository (Accession #GSE136674). All other

## Abstract

Productive transmission of malaria parasites hinges upon the execution of key transcriptional and posttranscriptional regulatory events. While much is now known about how specific transcription factors activate or repress sexual commitment programs, far less is known about the production of a preferred mRNA homeostasis following commitment and through the host-to-vector transmission event. Here, we show that in *Plasmodium* parasites, the NOT1 scaffold protein of the CAF1/CCR4/Not complex is duplicated, and one paralogue is dedicated for essential transmission functions. Moreover, this NOT1-G paralogue is central to the sex-specific functions previously associated with its interacting partners, as deletion of *not1-g* in *Plasmodium yoelii* leads to a comparable or complete arrest phenotype for both male and female parasites. We show that, consistent with its role in other eukaryotes, PyNOT1-G localizes to cytosolic puncta throughout much of the *Plasmodium* life cycle. PyNOT1-G is essential to both the complete maturation of male gametes and to the continued development of the fertilized zygote originating from female parasites. Comparative transcriptomics of wild-type and *pynot1-g⁻* parasites shows that loss of PyNOT1-G leads to transcript dysregulation preceding and during gametocytogenesis and shows that PyNOT1-G acts to preserve mRNAs that are critical to sexual and early mosquito stage development. Finally, we demonstrate that the tristetraprolin (TTP)-binding domain, which acts as the typical organization platform for RNA decay (TTP) and RNA preservation (ELAV/HuR) factors is dispensable for PyNOT1-G's essential blood stage functions but impacts host-to-vector transmission. Together, we conclude that a NOT1-G paralogue in *Plasmodium* fulfills the complex transmission requirements of both male and female parasites.

## Introduction

Malaria is still one of the great global health problems of the world today, with nearly 2 million new reported infections and 400,000 fatalities occurring annually [1]. Transmission of the

relevant data are within the paper and its Supporting Information files.

**Funding:** This work was supported by an NIAID R01AI123341 to SEL. The funders had no role in study design, data collection and analysis, decision to publish, or preparation of the manuscript.

**Competing interests:** The authors have declared that no competing interests exist.

**Abbreviations:** DIC, differential interference contrast; ELAV, embryonic lethal abnormal vision; FDR, false discovery rate; GO, gene ontology; IFA, immunofluorescence assay; IV, intravenous(ly; PCA, principal component analysis; SW, Swiss Webster; TIN, transcript integrity number; TTP, tristetraprolin; VST, variance-stabilizing transformation.

*Plasmodium* parasites that cause malaria occurs through the bite of a female anopheline mosquito, which transmit liver-infectious sporozoites to initiate a new infection of a mammalian host. Completion of the clinically silent liver stage of development results in the release of membrane-enclosed bundles called merosomes into the blood stream, which are filled with red blood cell–infectious merozoites that can be released upon passage through microcapillaries [2,3]. Merozoites initiate the asexual blood stage of development, thus initiating a rapid expansion of parasite numbers in the host. However, these asexual blood stage parasites cannot be productively transmitted back to the mosquito. For this, a fraction of blood stage parasites will commit to a sexual developmental program to produce male and female gametocytes, which is dependent upon the ApiAP2-G-specific transcription factor [4–6] and is also affected by ApiAP2-G2, which is thought to act as a transcriptional repressor of off-stage transcription [7–9]. In addition, several other protein- and epigenetic-based regulators are central to the commitment to sexual stage development (reviewed in [10,11]).

In contrast, less is known about the effects upon RNA metabolism and translational control during and following the commitment step of *Plasmodium*, which presumably would be regulated by RNA-binding proteins and their binding partners [12]. What is clear is that *Plasmodium* has evolved and adapted common eukaryotic mechanisms for translational repression to promote the efficiency of its transmission, which has been observed in *Plasmodium falciparum*, *Plasmodium berghei*, *Plasmodium yoelii*, and *Plasmodium vivax* (reviewed in [13]). Among the proteins implicated in *Plasmodium*'s translational control of specific mRNAs in female gametocytes are the following: DOZI (orthologue of an mRNA-decapping activator human DDX6/yeast Dhh1), CITH (orthologue of Lsm14A), PUF family proteins (PUF1 and PUF2), and ALBA family proteins (ALBA1-4) [14–20]. In addition, the CAF1 and CCR4-1 deadenylase members of the CAF1/CCR4/NOT complex also appear to play an important role in male gametocyte development and in host-to-vector transmission by counterintuitively helping to preserve mRNAs, although the mechanism by which this occurs (e.g., stabilization) remains to be demonstrated [21].

In model eukaryotes and human cells, the CAF1/CCR4/NOT complex is best known for its role as the major mRNA deadenylation complex of eukaryotes, but it touches upon many aspects of RNA metabolism from the birth to death of mRNAs, as well as translational repression of specific transcripts (reviewed in [22]). The structure of the complex was solved from *Schizosaccharomyces pombe*, which confirmed the modular nature of this complex with effector proteins being recruited to a central NOT1 scaffold [23]. Together with higher-resolution structures of different interaction modules and interacting proteins from several eukaryotes (NOT1 with DDX6, the NOT Module (NOT1-NOT2-NOT5), NOT1 C-terminal domain-NOT4, NOT1 with CAF40, CAF1, CCR4; reviewed in [24]), we have a reasonably good understanding of the overall composition and architecture of this complex. Moreover, the functions of this complex are well known as well. The decay of mRNAs first typically involves the shortening of the poly(A) tail by CAF1/CCR4/NOT and/or Pan2/Pan3 (although Pan2/Pan3 have not been identified in Apicomplexa). This is then followed by the removal of the $m^7G$ 5′ cap by DCP1/DCP2, which can be activated by a DDX6 protein [25,26]. Finally, the now unprotected mRNA is degraded by 5′ to 3′ and 3′ to 5′ exonucleases.

In these processes, NOT1 acts as a central nexus for the overall functions of its complex. Because NOT1 acts as an essential scaffold and yet lacks enzymatic activity of its own, recruitment of the appropriate effector proteins and their regulators is critical for the function of this complex. In addition to the proteins noted above, embryonic lethal abnormal vision (ELAV) family proteins and tristetraprolin (TTP) are recruited and act as antagonistic, mutually exclusive modulators to functionally toggle the complex's activity between transcript preservation versus transcript degradation [27]. ELAV family proteins (which include HuR and CELF

proteins) and TTPs (which are C3H1 zinc finger proteins and include ZFP36, TIS11) are RNA-binding proteins that bind with AU-rich elements in eukaryotes [28]. In addition to its roles in RNA metabolism, recently, it was shown that the CAF1/CCR4/NOT complex acts to monitor for low translational efficiency. In this work, NOT5 was found to associate with ribosomes attempting to translate suboptimal codons by binding the ribosomal E-site in its post-translocation state [29]. Those authors proposed that the association of CAF1/CCR4/NOT and Dhh1 (the orthologue of DDX6, DOZI) with these ribosomes prompts cotranslational decay of these mRNAs with an abundance of suboptimal codons via deadenylation and decapping. As the localization of Dhh1 and orthologues to cytosolic processing bodies is associated with its activity, it is notable that the association of NOT1 with Dhh1 acts to inhibit the assembly of processing bodies in yeast [30]. Taken together, as NOT1 itself lacks any enzymatic activities, it brings together protein effectors that can degrade or preserve mRNAs and can selectively target mRNAs that are not optimally coded for translation. Due to key differences between *Plasmodium* and other eukaryotes, such as its extreme AU-rich transcriptome, it is not certain if the same activities are used in the same ways in *Plasmodium* parasites.

Previous work that aimed to determine whether *Plasmodium* uses the CAF1/CCR4/NOT complex similarly was first focused on the *P. falciparum* CAF1 deadenylase, which was identified through the *piggyBac* transposon screen [31]. While initially thought to be dispensable, further experiments demonstrated that only the conserved N-terminal CAF domain was essential and remained expressed with the *piggyBac* insertion [21]. We have recently shown that both the specialty CCR4-1 deadenylase and the generalist CAF1 deadenylase are critical for RNA metabolism in gametocytes and affect the synchronicity of male gametocytogenesis and transmission to mosquitoes [21]. In this work, we experimentally identified that the CCR4-1 deadenylase interacts with canonical members of the CAF1/CCR4/NOT complex (NOT2, NOT4, NOT5, CAF40), including both deadenylases (CCR4-1, CAF1), as well as the *Plasmodium* orthologue of DDX6 (DOZI), CITH, an ELAV family protein (CELF2/Bruno/HoBo), and more. Moreover, we also identified ribosomal proteins, suggesting that a physical link between CAF1/CCR4/NOT via NOT5 could be present in *Plasmodium* as has been seen for model eukaryotes [29]. Through label-free quantification, we also identified a highly abundant, but uncharacterized protein annotated simply as "NOT Family Protein."

Here, we demonstrate that in contrast to other sequenced eukaryotes, *Plasmodium* and 2 related parasites have duplicated the *not1* gene, and *Plasmodium* has dedicated one of the paralogues for essential transmission-related functions. We have reannotated these paralogues as NOT1 and NOT1-G according to their phenotypes and functions. We show that NOT1-G is responsible first for a slight dampening of the production of gametocytes and yet then promotes the maturation and fertility of those gametocytes that do commit. PyNOT1-G affects both sexes, as it is essential for the production of male gametes and the continued development of fertilized female gametes in early mosquito stage development. Because the CAF1/CCR4/NOT complex associates with members of the DOZI/CITH/ALBA complex in *P. yoelii*, these phenotypes indicate that PyNOT1-G is a central and essential organizer of gametocyte development.

## Results

### Parasites of the Aconoidasida class have duplicated the *not1* gene

Control of mRNA metabolism is a central feature of eukaryotic gene regulation. The CAF1/CCR4/NOT complex plays many central roles in these processes from the birth to the death of mRNAs, with its best appreciated role being in the decay of the poly(A) tail of mRNAs. Previously, we and others have demonstrated the importance of the CAF1 and CCR4-1

deadenylases of this complex to the sexual development of both rodent-infectious and human-infectious malaria parasites (*P. yoelii* and *P. falciparum*, respectively) [21,31]. In our previous efforts to experimentally determine the composition of the CAF1/CCR4/NOT complex in *Plasmodium* through mass spectrometry–based proteomics, we identified a protein annotated only as "NOT Family Protein" (PY17X_1027900) that was found to be associated with CCR4-1 [21]. Bioinformatic analyses using PlasmoDB and BLASTp identified that the protein encoded by PY17X_1027900 mostly closely matched eukaryotic NOT1 proteins but lacked a bioinformatically predictable TTP-binding domain that is present in the gene currently annotated as NOT1 (PY17X_0945600) (Fig 1) [32,33]. Because no other sequenced eukaryotes to date have an annotated duplication of *not1*, we used bioinformatic sequence comparisons to explore this further. We concluded that these 2 *not1* genes were paralogues due to the high degree of sequence conservation in specific domains at both the DNA and protein levels (DNA: CAF1-binding domain 72% identity, NOT4-binding domain 66% identity; Protein: CAF1-binding

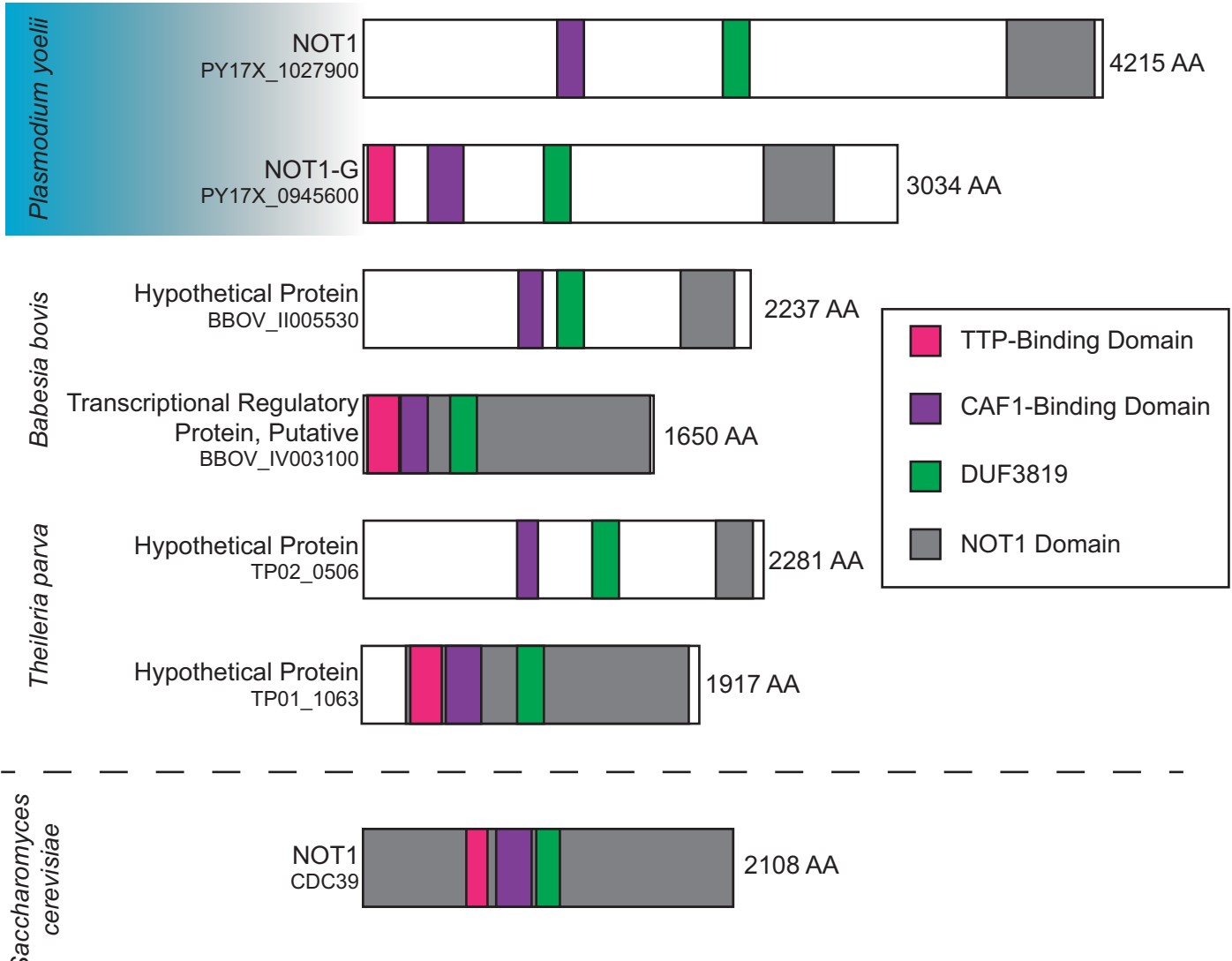

**Fig 1. NOT1 is duplicated in the Aconoidasida class of apicomplexans.** Shown here are examples of NOT1 proteins from the Aconoidasida class of apicomplexans with a representative model eukaryote NOT1 protein from *Saccharomyces cerevisiae*. Protein and domain sizes are drawn to scale. Bioinformatically predicted domains were identified through BlastP searches and amino acid alignments. TTP, tristetraprolin.

domain 41% identical/63% positive, CAF40-binding domain 33%/57%, NOT4-binding domain 48%/62%, NOT1 domain 50%/73%). Moreover, these genes are highly conserved, syntenic, and only present across *Plasmodium* species and 2 other species of the Aconoidasida class (Genera: *Theileria*, *Babesia*) but not in other apicomplexans, model eukaryotes, or humans. Together, this provides evidence that these are paralogues of NOT1 that arose from a gene duplication in the common evolutionary ancestor [34]. As these parasites share some common traits for their transmission strategies between mammals and insects that are not shared with other apicomplexans, we hypothesized that this NOT1 paralogue may have evolved for specialized functions during parasite transmission.

### *Plasmodium*'s NOT1 paralogues localize to cytosolic puncta

Due to the central roles that the CAF1/CCR4/NOT complex plays in RNA metabolism in eukaryotes, this complex is typically found in the nucleus and/or in cytosolic puncta associated with complexes in the spectrum between stress granules and processing bodies (P-bodies) [22,35]. Representative members of this complex (CAF1, CCR4-1) in *Plasmodium* also have this expression and localization profile and are found predominantly in cytosolic puncta and associate with proteins that are commonly associated with these granules [21]. To determine if the 2 paralogues of NOT1 are used at different points in the *Plasmodium* life cycle, we used conventional reverse genetics approaches to append a C-terminal GFP tag to each NOT1 paralogue in *P. yoelii* (S2 Fig) and assessed their expression and localization (Figs 2 and S2). Both NOT1 paralogues were expressed and localized at the same times in the life cycle, and the

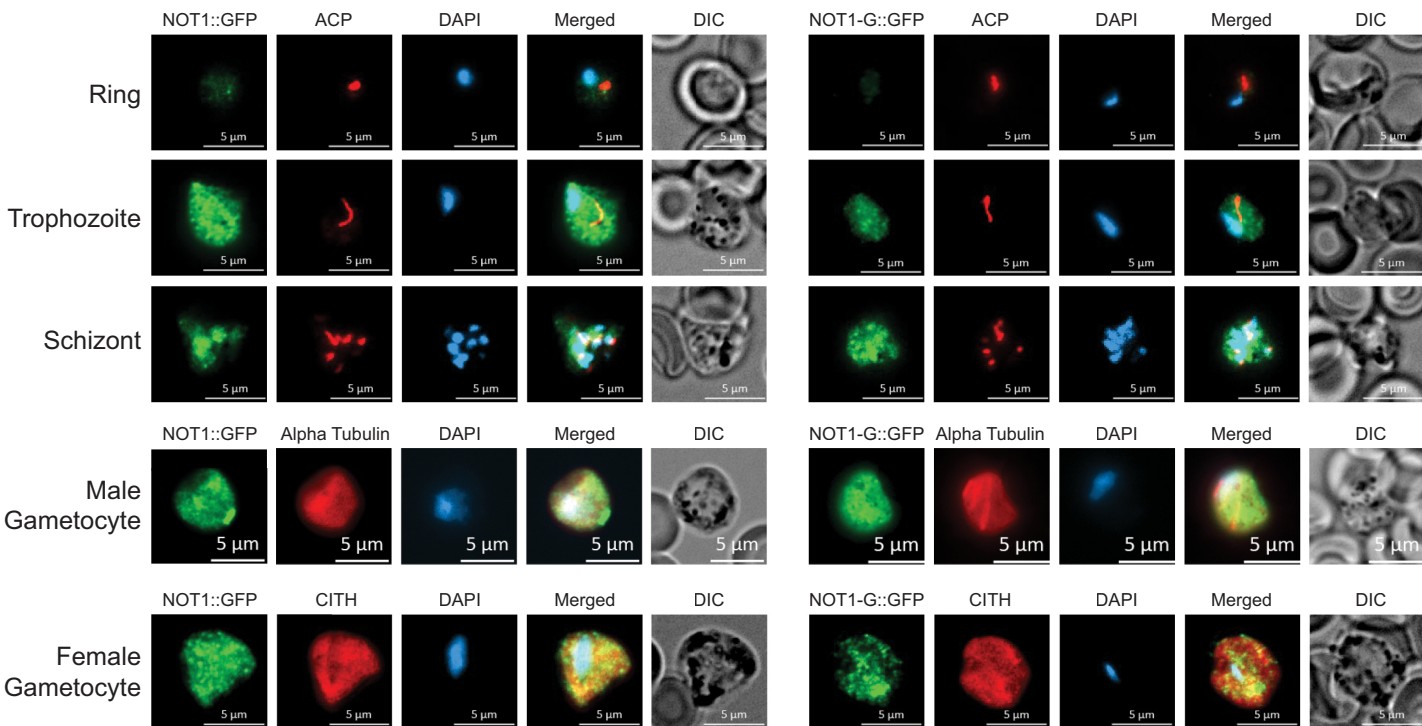

**Fig 2. PyNOT1 and PyNOT1-G have similar expression patterns and localize to cytosolic puncta.** Representative IFA images of (A) PyNOT1::GFP and (B) PyNOT1-G::GFP blood stage parasites show localization patterns typical for the CAF1/CCR4/NOT complex in eukaryotes. All samples were stained with DAPI and mouse anti-GFP to image PyNOT1::GFP or PyNOT1-G::GFP. Asexual blood stage parasites were counterstained with anti-PyACP and DAPI to identify ring, trophozoite, and schizont stages. Gametocytes were counterstained with anti-alpha tubulin (male) or anti-PyCITH (female). Scale bar is 5 μm. DIC, differential interference contrast; IFA, immunofluorescence assay.

expression pattern for PyNOT1 and PyNOT1-G is cytosolic and at least partially punctate. This is consistent with the localization of other members of the CAF1/CCR4/NOT complex in *Plasmodium*, model eukaryotes, and human cells [21,22]. Of note, the NOT1 paralogues were expressed in both asexual and sexual blood stage parasites with a nonuniform, cytosolic distribution with some puncta visible. Both paralogous proteins were expressed in mosquito stage parasites, with localization shifting from a cytosolic diffuse pattern in oocysts, to a more nucleus-proximal pattern in oocyst sporozoites, and, finally, to a more apical pattern in salivary gland sporozoites. No expression of either protein was observed in mid- or late-liver stage parasites. Together, these expression and staining patterns match what is commonly seen for members of the CAF1/CCR4/NOT complex in other eukaryotes, as well as what we observed previously in *P. yoelii* with 2 other members of this complex: CAF1 and CCR4-1 [21]. Coupled with proteomic data showing their association with CCR4-1, this strongly indicates that these NOT1 paralogues are resident members of the CAF1/CCR4/NOT complex in *P. yoelii*.

## PyNOT1 (PY17X_1027900) is critical for asexual blood stage growth in *P. yoelii*

NOT1 is essential and acts as a central scaffold to nucleate several RNA-centric functions in eukaryotes [22]. The CAF1/CCR4/NOT complex is most well known for its role in mRNA deadenylation but also contributes to transcriptional initiation and elongation, translational repression, mRNA export, and nucleus-based quality control activities. To determine if either *Plasmodium* NOT1 paralogue bore a greater functional semblance to canonical NOT1 proteins in eukaryotes or had dedicated functions in *Plasmodium*, we used conventional reverse genetics approaches to delete the coding sequence of each paralogue, which was confirmed by genotyping PCR (S3 Fig).

In agreement with the critical/essential role of NOT1 to eukaryotic RNA metabolism, our attempts to delete *py17x_1027900* were largely unsuccessful. However, in 1 of 2 technical duplicates of 6 independent transfection attempts, we were able to obtain a 100% transgenic population with an extremely slow growth phenotype. It was only possible to produce this parasite line because the population was entirely transgenic, as the presence of even a small number of wild-type parasites with de novo resistance to pyrimethamine would likely have rapidly outgrown these transgenic parasites in the mouse. These data align with results from both PlasmoGEM (*P. berghei*) and *piggyBac* (*P. falciparum*) genetic screens that noted the importance of this gene to asexual blood stage growth [36–38]. Due to the severe asexual blood stage defect, which closely aligned with observations of NOT1-related phenotypes in other eukaryotes, we propose that this "NOT Family Protein" is truly the NOT1 protein of *Plasmodium* parasites. Finally, it is notable that in contrast to other eukaryotes, PyNOT1 can be deleted and indicates that it is not strictly essential.

## PyNOT1-G (PY17X_0945600) dampens sexual stage commitment but is essential for sexual maturation

In stark contrast, the paralogous gene currently annotated as NOT1 (*py17x_0945600*) was readily deleted, and multiple independent clones were obtained by limited dilution cloning (S3B Fig). This was unexpected given the essential role of NOT1 in all other eukaryotes and lends further credence to the assignment of the first paralogue described above as the true NOT1 protein of *Plasmodium* parasites. The ability to disrupt this gene was corroborated by the *piggyBac* screen in *P. falciparum*, which identified that transposon insertions in this gene could still produce viable parasites [36].

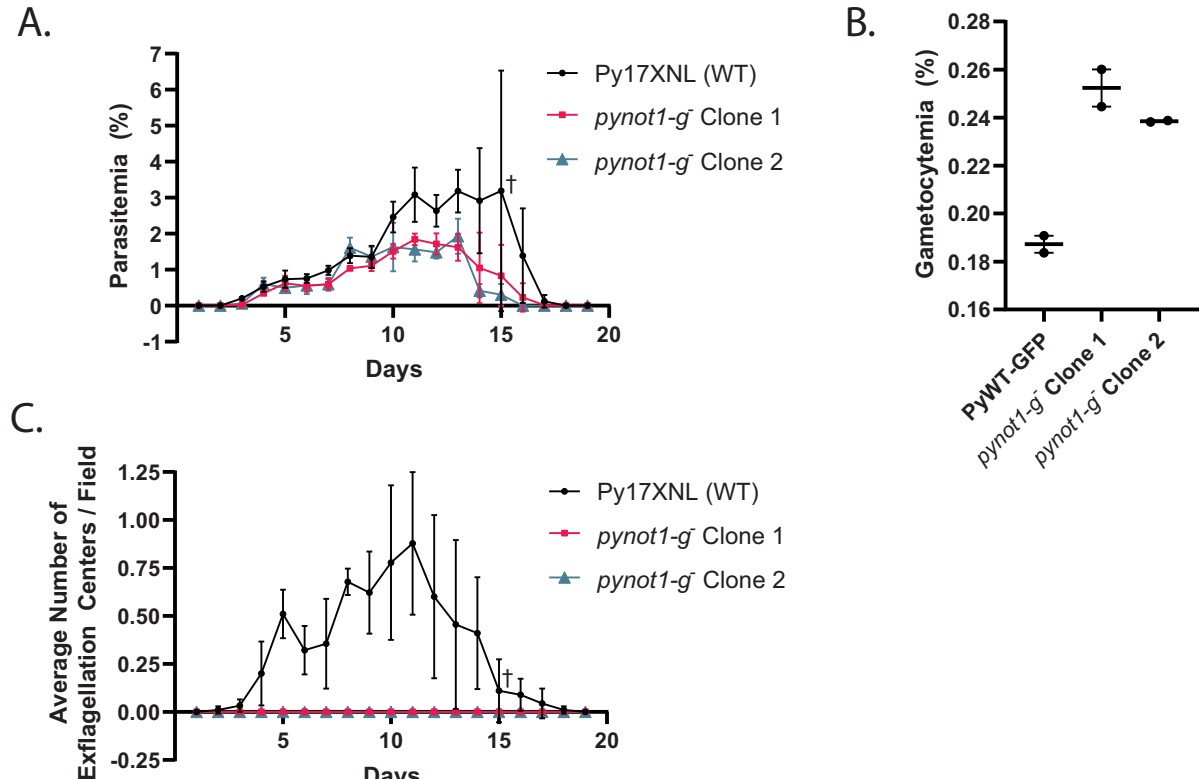

**Fig 3. PyNOT1-G is essential to the formation of male gametes.** (A) The blood stage growth of Py17XNL wild-type parasites and 2 independent clones of *pynot1-g⁻* parasites was compared over the entire course of infection in biological triplicate, with each replicate done in technical triplicate. Maximal parasitemia of *pynot1-g⁻* parasites was lower and resulted in slightly faster clearance by the mouse. (B) The number of gametocytes produced by Py17XNL WT-GFP parasites and 2 independent clones of *pynot1-g⁻* parasites was measured by flow cytometry using the presence of GFP fluorescence in biological duplicate, with each replicate done in technical triplicate. Average values for each biological replicate, their median, and range are provided. Statistical comparisons used an unpaired *t* test with Welch correction. (C) Male gametogenesis was measured by DIC microscopy to count the number of exflagellation centers ("centers of movement") per microscopic field using a 10× eyepiece and 40× objective lens. The same mice and time points were used as in panel A and were assessed in biological triplicate, with each replicate done in technical triplicate. A cross symbol (A, C) indicates that a single mouse infected with Py17XNL wild-type parasites was euthanized on day 15 due to parasitemia more than 10% as required by our approved vertebrate animal protocols. Error bars indicate the standard deviation. Underlying data are provided in S1 Data. DIC, differential interference contrast.

In order to determine if *py17x_0945600* plays a specialized role in *Plasmodium* growth, development, or transmission, mice were infected with 10,000 Py17XNL wild-type parasites or clonal *py17x_0945600*-null parasites, and the progression of the asexual blood stage infection was monitored by daily Giemsa-stained thin blood smears in technical and biological triplicate. No significant asexual blood stage growth defects were observed for *py17x_0945600*-null parasites until late in the blood stage infection (day 10). At this point, there was a small but statistically significant decrease in peak parasitemia with a concomitant earlier time to parasite clearance (Fig 3A).

One explanation for this defect could be due to an overcommitment to gametocytogenesis in the absence of this gene product, as committed male and female parasites are differentiated cells and cannot contribute to increasing the total parasitemia further. Moreover, this asexual blood stage growth defect became pronounced approximately when gametocyte numbers begin to be abundant in the *P. yoelii* 17XNL strain (approximately 1% parasitemia). To assess this, we compared the total gametocytemia of Py17XNL WT-GFP parasites and *py17x_0945600*-null parasites following treatment with a 2-day course of sulfadiazine to

   

selectively kill asexual blood stage parasites and to leave gametocytes in circulation. By a flow cytometric assay to count GFP-expressing parasites, we observed a slight increase in gametocytogenesis (27% to 35% increase, unpaired *t* test with Welch correction, $p = 0.04$) in *py17x_0945600*-null parasites compared to wild-type parasites (Fig 3B and S2 File). The increase in gametocytemia paired with the decrease in asexual blood stage parasitemia may indicate that PY17X_0945600's role in wild-type parasites is to slightly dampen commitment to gametocytogenesis. To assess this effect upon gametocytes further, the same mice that were used to assess the asexual blood stage growth kinetics were also monitored for the presence of male gametes through observation of exflagellation centers ("centers of movement") (Fig 3C). We observed no exflagellating male gametes throughout the entire course of blood stage infection, indicating that PY17X_0945600 is essential for either gametocyte maturation and/or gametogenesis. Taken together, we have reannotated PY17X_0945600 as PyNOT1-G due to its dual and opposed functions to perhaps dampen gametocyte commitment and yet act in an essential manner to drive male gametocyte maturation.

## PyNOT1-G is essential to zygote development

While these data indicate the PyNOT1-G is essential for the complete maturation of male parasites, these experiments did not directly study the impact of PyNOT1-G on female gametocytes or upon host-to-vector transmission. To address these questions, we conducted a genetic cross experiment with Py17XNL wild-type parasites that do not express a fluorescent protein and *pynot1-g⁻* parasites that express GFPmut2 from a constitutive promoter integrated in the *pynot1-g* locus. If either male or female *pynot1-g⁻* parasites were viable for transmission either by self-fertilization or by crossing with wild-type parasites, GFP-expressing oocysts would be evident in the mosquito (Fig 4A). In contrast, if only wild-type male and female parasites were fertile, only nonfluorescent oocysts would be observed. In 3 independent transmission experiments, donor mice infected with either Py17XNL wild-type parasites or *pynot1-g⁻* parasites were used to produce parasites for IV transfer of (1) 10,000 wild-type parasites; (2) 5,000 wild-type parasites + 5,000 *pynot1-g⁻* parasites; or (3) 10,000 *pynot1-g⁻* parasites into experimental mice. Parasitemia increased to 1% with the same timing in all mice and exflagellating male gametes were observed in mice infected with wild-type parasites or a combination of wild-type and *pynot1-g⁻* parasites but were not seen with infections with *pynot1-g⁻* parasites. Mice were anesthetized and mosquitoes were allowed to blood feed for 1 period of 15 minutes. Seven days post-blood meal, midguts were removed and oocysts were counted and scored by differential interference contrast (DIC) and fluorescence microscopy. No oocysts were detected in mosquitoes that fed upon mice infected with *pynot1-g⁻* parasites, which is consistent with the absence of exflagellating male *pynot1-g⁻* gametes (Fig 3C). However, in mosquitoes that fed upon wild-type parasites or a mixture of wild-type and *pynot1-g⁻* parasites, only nonfluorescent oocysts were detected, thus indicating that both male and female *pynot1-g⁻* parasites are unable to productively transmit to mosquitoes. Additionally, the effect upon both sexes of *pynot1-g⁻* parasites was further corroborated by the presence of approximately half of the number of oocysts in mosquitoes that fed upon the mixture of wild-type and *pynot1-g⁻* parasites as compared to those that fed upon only wild-type parasites. This indicates that PyNOT1-G is an essential driver of maturation for both male and female *Plasmodium* parasites.

As these defects in *pynot1-g⁻* parasites parasite transmission could result from a block in early mosquito stage development of female parasites, we used an in vitro genetic cross approach to determine if either sex of *pynot1-g⁻* parasites could produce zygotes or ookinetes. As controls, we used PyWT-mScarlet (Py1115) and PyWT-GFP (Py489) transgenic

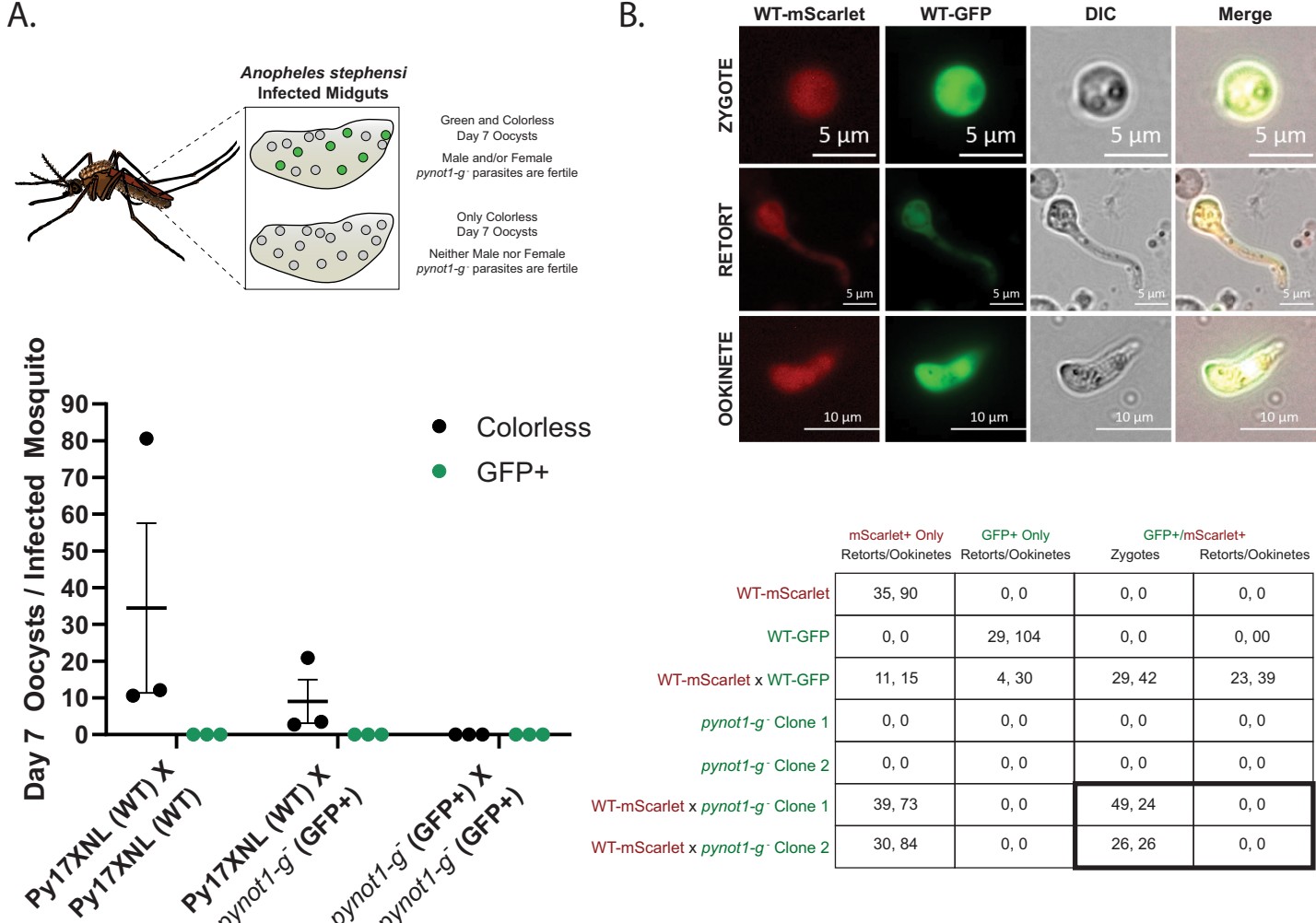

**Fig 4. PyNOT1-G is essential to the development of zygotes resulting from female gametocytes.** (A) A genetic cross experiment with Py17XNL wild-type parasites (colorless) and *pynot1-g⁻* (GFP+) transgenic parasites was conducted to assess effects upon parasite transmission. Mice were infected with a total of 10,000 mixed blood stage parasites consisting of only wild-type parasites, an equal mixture of wild-type and transgenic parasites, or only transgenic parasites. Mosquito transmission occurred on the day where peak exflagellation was observed for mice infected with only wild-type parasites, and colorless and GFP+ oocysts were counted 7 days post-blood meal by fluorescence and DIC microscopy. An illustration of the experimental readout/interpretation (top) and the results (bottom) are provided. Experiments were conducted in biological triplicate, with error bars denoting SEM. (B) A genetic cross experiment with Py17XNL WT-mScarlet, WT-GFP, and *pynot1-g⁻* (GFP+) transgenic parasites was conducted using in vitro ookinete culture conditions. Equal volumes of infected blood at comparable parasitemias were mixed as indicated and were scored by live fluorescence microscopy for the presence/absence of mScarlet and GFP signals. Representative images of the WT-mScarlet x WT-GFP genetic cross are provided to show zygote, retort, and ookinete stages (top). Results of all tested genetic crosses are provided (bottom) from experiments conducted in biological duplicate. Parasites resulting from the WT-mScarlet x *pynot1-g⁻* (GFP+) cross could not develop beyond the zygote stage. Scale bars are 5 μm (zygote, retort) or 10 μm (ookinete). Underlying data are provided in S1 Data. DIC, differential interference contrast.

lines. As anticipated, we found that wild-type parasites could readily self-cross (GFP+/mScarlet−, GFP−/mScarlet+) and cross (GFP+/mScarlet+) to yield zygotes, retorts, and ookinetes (Fig 4B). In contrast, experiments that crossed PyWT-mScarlet with GFP-expressing *pynot1-g⁻* parasites could not yield *pynot1-g⁻* self-crossed zygotes but could yield crossed (GFP+/mScarlet+) zygotes that failed to develop further (Fig 4B). This female-specific phenotype matches that seen previously for *pbdozi⁻* and *pbcith⁻* parasites and indicates that PyNOT1-G may play an important role in preparing the female gametocyte for further development [19,20].

## Extensive transcriptomic dysregulation in *pynot1-g⁻* schizonts and gametocytes

As the CAF1/CCR4/NOT complex can contribute toward both mRNA preservation and mRNA decay activities in *Plasmodium* and other eukaryotes [21,22], we used total comparative RNA-seq (WT versus *pynot1-g⁻* parasites) to assess differences in transcript abundances in both mixed male and female gametocyte populations and asexual blood stage schizonts depleted of rings, trophozoites, and gametocytes through a subtractive magnetic approach. A variance-stabilizing transformation (VST) of the sample raw counts was used to generate a principal component analysis (PCA) plot of these data, which shows good within-group clustering and separation of different sample types (Fig 5A). A heatmap of the expression of 50 selected transcripts for gametocytes and schizonts illustrates this clustering and changes in gene expression as well (Fig 5B). MA plots and Volcano plots of gametocytes (Fig 5C and 5E) and schizonts (Fig 5D and 5F) show differential expression of specific transcripts in both stages that is dependent upon PyNOT1-G.

First, we further examined the effect of PyNOT1-G in gametocytogenesis through these comparative RNA-seq data of Py17XNL wild-type and *pynot1-g⁻* gametocytes (y5C and 5E and S1 Table). In addition to the use of DEseq2, we have also imposed a transcript integrity number (TIN) metric to help identify and remove reads due to run-in transcription [39,40]. We observed that *pynot1-g⁻* gametocytes had lower abundances of 427 transcripts (threshold of $\leq \log_2 -2.5$, false discovery rate (FDR) $< 0.05$), including a substantial number of gametocyte-enriched (*p25*, *p28*, 3 *apiap2* transcripts, *imc*-related transcripts, *ccp* family transcripts, *cpw-wpc* family transcripts, *migs*, and *actin II*) and early mosquito stage (*ctrp*, *warp*, *psop* family transcripts, and *perforins*) transcripts [4,41]. Notably, the transcript abundance of the NOT1 paralogue (*pynot1*) was not affected, whereas *pyapiap2-g*, the master regulator of commitment to gametocytogenesis, was down $\log_2 -1.2$ (2.3-fold reduction) in *pynot1-g⁻* parasites. Enriched gene ontology (GO) terms include cellular component terms related to the cytoskeleton, microtubules/actin, and molecular function terms related to binding to the cytoskeleton (S1 Table). CirGO plots of the GO terms (biological processes) associated with the differentially expressed transcripts illustrate this further, with several specific terms present that are related to these same processes, structures, and features for both gametocytes and asexual blood stage schizonts (Fig 5G and 5H). In contrast, few transcripts had increases in abundance (threshold of $\geq \log_2 2.5$, FDR $< 0.05$), but those that were more abundant are from off-stage genes related to invasive stages and/or sporozoites. Further expansion of the threshold ($\geq \log_2 2.0$, FDR $< 0.05$) then included many more off-stage, apical organelle gene products (MSP family, RON family, RhopH family, RAP family, SERA family, MAEBL, TREP).

This transcriptional dysregulation is catastrophic for male gametocytes, as they are unable to complete gametogenesis. However, despite this substantial dysregulation of transcript abundances, it is remarkable that female gametocytes are still permissive to form female gametes, which can be fertilized by competent wild-type male gametes. However, because developmental arrest occurs at the zygote stage, this indicates that the RNA homeostasis promoted by the PyNOT1-G complex is essential for further development, as was seen for studies of PbDOZI and PbCITH [19,20]. In agreement with this, we compared differentially expressed transcripts from *pynot1-g⁻* and *pbdozi⁻* and/or *pbcith⁻* parasites and found that 98 of the 115 transcripts that are $\leq \log_2 -1$ ($p < 0.05$) in *pbdozi⁻* and/or *pbcith⁻* are also reduced in abundance $\leq \log_2 -2.5$ in *pynot1-g⁻* parasites. By matching the reporting thresholds to $\leq \log_2 -1$ ($p < 0.05$), nearly all (105/115) of the transcripts that are lower in abundance in *pbdozi⁻* and/or *pbcith⁻* parasites are also in lower abundance in *pynot1-g⁻* parasites. This highly similar effect may be attributable to these proteins working in concert, as DOZI and CITH are known to interact

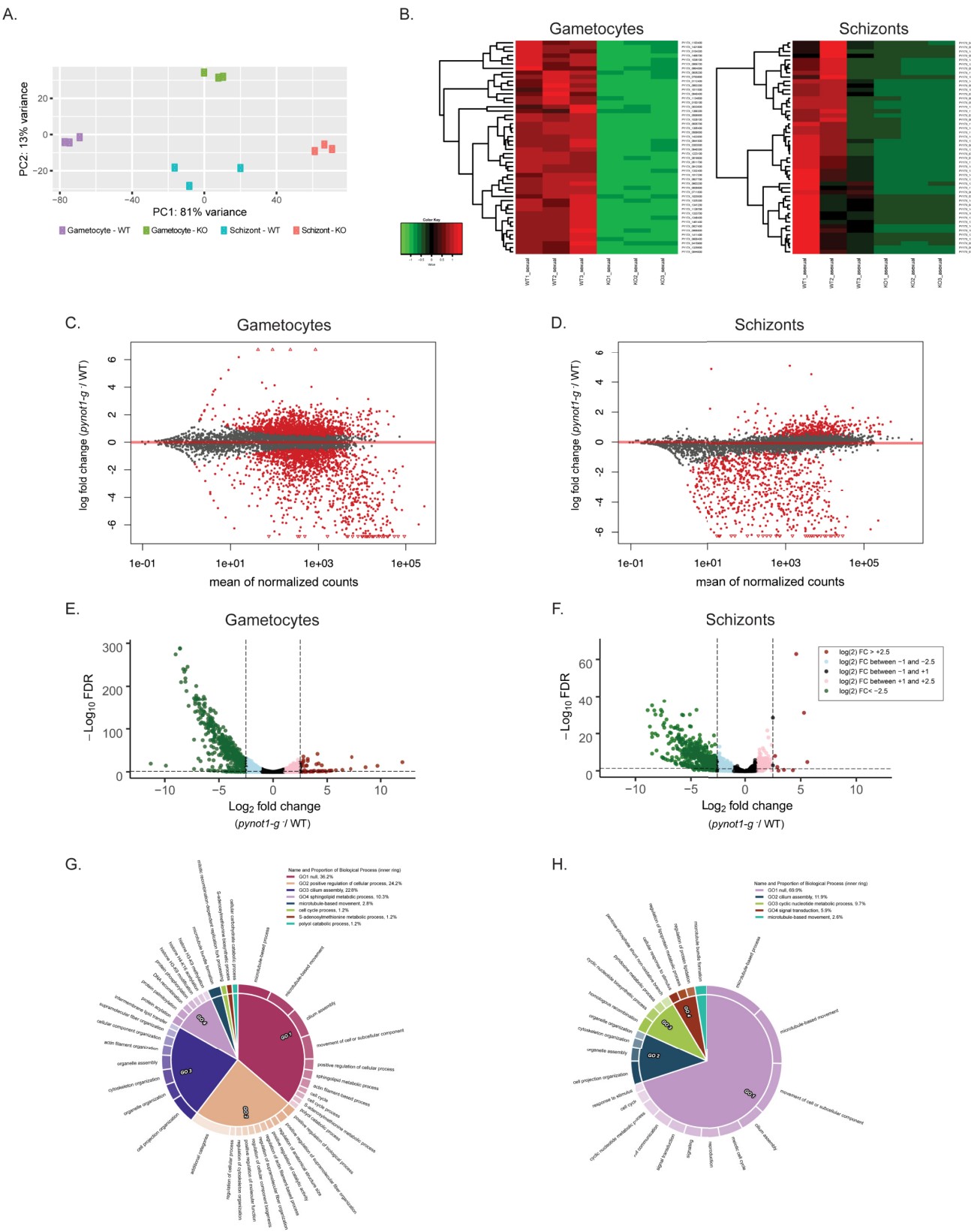

**Fig 5. Extensive dysregulation of mRNAs in *pynot1-g⁻* schizonts and gametocytes.** Comparative RNA-seq of wild-type or *pynot1-g⁻* transgenic parasites was conducted using mixed male and female gametocytes or schizonts that were enriched through a subtractive magnetic depletion approach. (A) VST-transformed raw counts were used as input values to generate a PCA plot to demonstrate sample clustering. (B) Heatmaps of the expression of 50 selected transcripts for gametocytes (left) and schizonts (right) further illustrate clustering and the effect of deleting *pynot1-g*. (C and D) MA plots generated from the estimated shrunken $\log_2$ fold changes are shown for gametocytes (C) and schizonts (D), with data points shaded in red indicating $p < 0.05$ (comparing *pynot1-g⁻*/WT). Data points that do not meet this threshold are denoted as open triangles (oriented up or down). (E and F) Plotted are the $-\log_{10}$ FDR vs the $\log_2$ fold change values for differentially expressed transcripts as per DEseq2, with transcripts shaded based upon their fold change values (comparing *pynot1-g⁻*/WT). Similar gametocyte- and early mosquito stage-enriched transcripts were significantly lower in abundance in both stages. (G and H) CirGO plots of GO terms for gametocytes (left) and schizonts (right) are provided that highlight key biological processes that are impacted by PyNOT1-G. FDR, false discovery rate; GO, gene ontology; PCA, principal component analysis; VST, variance-stabilizing transformation.

with the CAF1/CCR4/NOT complex in *Plasmodium* and other eukaryotes [21,22]. Additionally, many of the same transcripts that are similarly dysregulated in the *pyalba4⁻* line are also dysregulated in *pynot1-g⁻* parasites (265 of 438 transcripts are $\leq \log_2 -1$ ($p < 0.05$) in both lines) [14]. Finally, 81 transcripts are similarly dysregulated across all 4 transgenic parasite lines (*pbdozi⁻*, p*bcith⁻*, *pynot1-g⁻*, and *pyalba4⁻*) and include *p25*, *lccl* family, IMC-related, *ap2-o*, *warp*, and *psop* family gene products (S2 Table). These results also have extensive overlap with PbDOZI-associated transcripts identified by RIP-chip (409 of 1,046 transcripts) [16].

However, as we also determined that PyNOT1-G has an impact on the peak and clearance of asexual blood stage growth, we also assessed changes in transcript abundance in asexual blood stage schizonts that can already be in the process of commitment. As blood stage infections of *P. berghei* and *P. yoelii* often contain a significant number of gametocytes (10% of total parasites), we developed a magnet-based subtractive separation approach to remove trophozoites, schizonts, and gametocytes from the sample through 4 serial passes over magnetized columns. Flow cytometric analyses demonstrated that gametocytes consisted of <1% of the remaining parasites and that additional passes through the magnetized columns provided no additional enrichment. The remaining ring stage parasites were cultured ex vivo until they reached a mature schizont stage based upon Giemsa staining and then were subjected to total comparative RNA-seq.

Here, we observed similar trends in the differential abundance of mRNAs that could be directly or indirectly preserved by PyNOT1-G and its complex in schizonts (Fig 5D and 5F and S3 Table). Those 293 mRNAs that are less abundant in *pynot1-g⁻* schizonts include many classic and recently defined early gametocyte transcripts [4,41]. These include many mRNAs that are translationally repressed in *P. berghei* and/or *P. falciparum*, such as *p25* and *p28* (49.5-fold and 74.6-fold reductions) as well as several members of the *psop*, *lccl*, and *ccp/lap* gene families. Moreover, other transcripts that were also dysregulated in gametocytes were similarly dysregulated in schizonts, including *warp*, *hsp20*, *migs*, *soap*, *p230p*, *p48/45*, *gest*, *gamete egress protein* (*gep*), *male development protein 1* (*mdv1*), *plasmepsin vIII*, and *actin ii*. Notably, *pyapiap2-g* is also down $\log_2 -2.3$ (approximately 5-fold) in schizonts. Statistically significant GO terms related to these transcripts, and to processes downstream of the schizont stage, include cellular component terms related to the crystalloid, cytoskeleton, microtubules, and dynein, molecular function terms related to motor activity, and biological process terms related to meiosis, microtubules, and movement. Together, this indicates that the specific transcripts that are regulated by PyNOT1-G in gametocytes are also affected by it prior to gametocytogenesis, suggesting that the setting of a preferred RNA homeostasis for host-to-vector transmission may occur earlier than anticipated. It remains to be determined if PyNOT1-G and its complex acts to stabilize mRNAs or if it acts in some other way to preserve the abundance of specific mRNAs. Importantly, these multifaceted roles of PyNOT1-G in RNA metabolism match and exceed the roles seen with CCR4-1 (one of the deadenylases of this complex) and the DOZI/CITH/ALBA4 that can associate with it [14,19–21].

## The tristetraprolin-binding domain of PyNOT1-G is dispensable for its essential roles in blood stage parasites but is important for transmission

The diverse functions of the CAF1/CCR4/NOT complex rely upon the associations of diverse regulators with the NOT1 scaffold to direct its activities to specific mRNAs. One such pair of antagonizing RNA-binding regulators, ELAVs (Hu family of proteins (HuR)) and TTP (C3H1 RNA-binding zinc finger proteins), bind to the same TTP-binding domain on NOT1 proteins and functionally toggle this complex between transcript preservation and transcript degradation, respectively [28]. While ELAV-like/HuR proteins can be bioinformatically predicted and have been found associated with the CAF1/CCR4/NOT and DOZI/CITH/ALBA regulatory complexes (CELF1, CELF2/Bruno/HoBo), a *Plasmodium* TTP orthologue cannot similarly be predicted with confidence [14,21,42]. The only bioinformatically discernable difference between the 2 NOT1 paralogues was the presence of a conserved TTP-binding domain (AA7-164) on NOT1-G, which is strongly conserved across *Plasmodium* species (e.g., 100% identical in *P. berghei* ANKA, 73% identical/88% similar in *P. falciparum* NF54, 80% identical/92% similar in *P. vivax*), but that cannot be detected on PyNOT1 or its orthologues in other *Plasmodium* species (Fig 1).

To this end, we used 2 complementary approaches to examine the importance of this putative TTP-binding domain to the function of PyNOT1-G. First, we created a transgenic parasite line using the strong, constitutive *pbeef1a* promoter to overexpress the TTP-binding domain (AA1-199) of PyNOT1-G fused to GFPmut2 (TTPbd::GFP) from a safe harbor locus (*pyp230p*) (S4A Fig). Expression boundaries for this protein variant were chosen based upon the use of domain predictions along with the presence of an enriched region of asparagine residues in *P. yoelii* that is even more pronounced in *P. falciparum*, as these regions are often found between functional domains in *Plasmodium* proteins. We hypothesized that overexpression of TTPbd::GFP would act dominant negatively by binding and sequestering proteins that need to interact with PyNOT1-G to be effective. Expression of TTPbd::GFP was observed by western blotting of the immunoprecipitated protein versus control WT-GFP parasites (S4B Fig). Despite this, we did not observe any statistically significant defects in asexual blood stage growth or the ability of these parasites to produce exflagellating male gametes (Fig 6A and 6B). Moreover, in stark contrast to *pynot1-g⁻* parasites, the TTPbd::GFP-overexpressing parasites were competent for transmission to mosquitoes (S4 Table). This indicates that overexpression of PyNOT1-G's TTP-binding domain does not act dominant negatively, perhaps because it could be insufficient to sponge away the relevant factors and/or because this domain and its interacting partners may not be essential to PyNOT1-G's functions in gametocytes.

Second, we created a transgenic parasite lacking the TTP-binding domain of PyNOT1-G by replacing those sequences with GFPmut2 (ΔTTPbd). Clonal lines were isolated and compared to Py17XNL WT-GFP parasites (S4C Fig). As with the TTPbd::GFP overexpression line, no statistically significant defect in asexual blood stage growth or in the numbers of exflagellating male gametes was detected (Fig 6C and 6D). However, statistically significant effects upon the prevalence of mosquito infection were observed (multiple unpaired *t* test, no assumptions regarding standard deviation, $p = 0.006763$ (WT versus Clone 1), $p = 0.030700$ (WT versus Clone 2)) but not for the number of oocysts per infected mosquito (Table 1). This indicates that the TTP-binding domain of PyNOT1-G, and by interference proteins that require it to interact with PyNOT1-G, are ultimately dispensable for PyNOT1-G's functions in male gametocytogenesis and/or gametogenesis and female parasite development beyond the zygote stage, but yet it appears to play a significant role in host-to-vector transmission. In contrast, the remaining portions of PyNOT1-G are essential for these functions and thus may provide additional interaction points for these regulatory proteins.

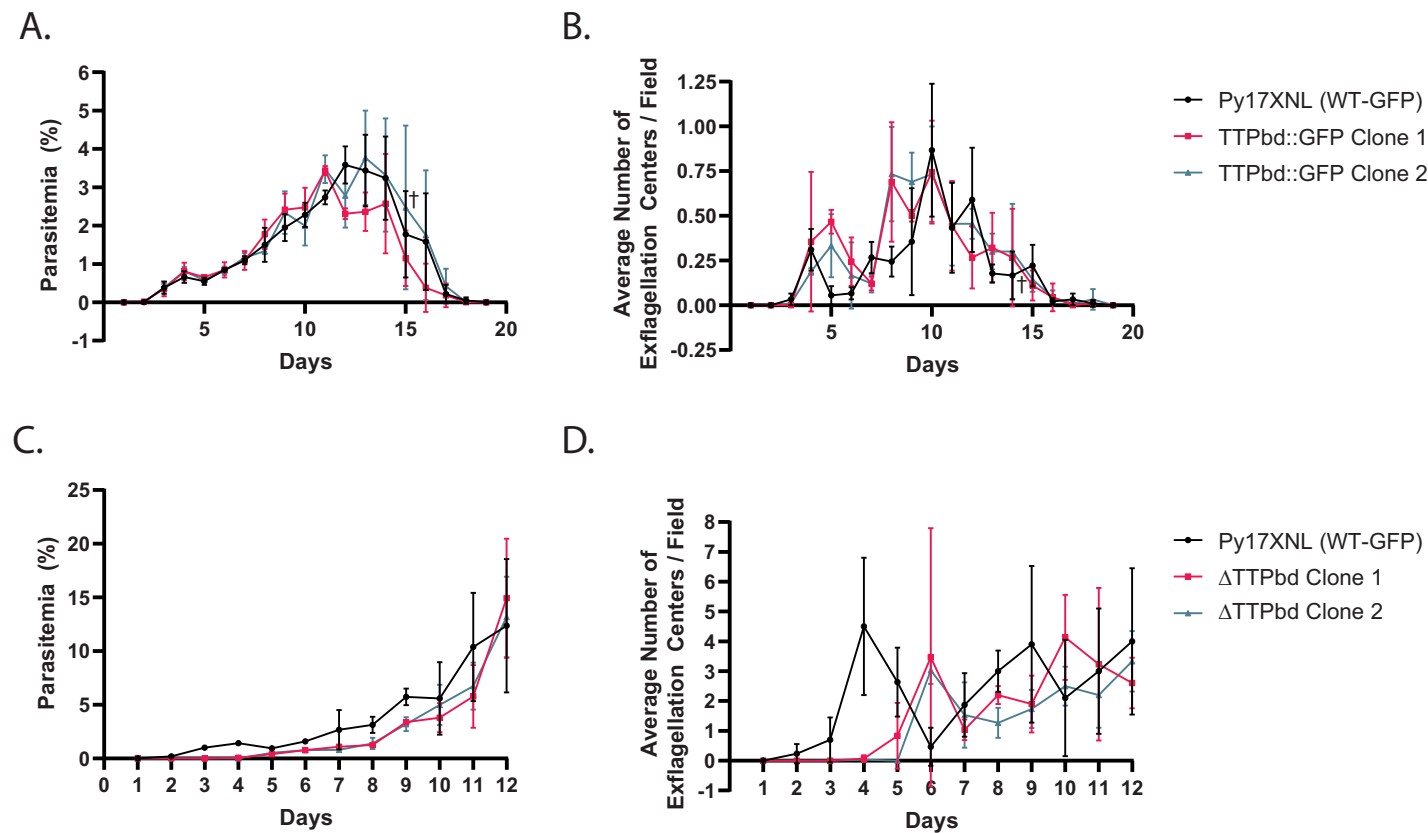

**Fig 6. The putative TTP-binding domain is dispensable for all essential blood stage functions of PyNOT1-G.** Clonal transgenic parasites overexpressing the predicted TTP-binding domain fused to GFP (TTPbd::GFP) from the safe harbor *p230p* genomic locus were compared to Py17XNL WT-GFP parasites for (A) asexual blood stage growth and (B) male gametogenesis. Clonal transgenic parasites expressing a variant of PyNOT1-G where the N-terminal TTPbd has been replaced with GFP were compared to Py17XNL WT-GFP parasites for (C) asexual blood stage growth and (D) male gametogenesis. A cross symbol (A, B) indicates that a single mouse infected with TTPbd::GFP Clone 2 parasites was euthanized on day 15 due to parasitemia more than 10% as required by our approved vertebrate animal protocols. Error bars indicate the standard deviation. Underlying data are provided in S1 Data. TTP, tristetraprolin.

## Discussion

Here, we describe the unique duplication and adaptation of a NOT1 paralogue for the transmission requirements of malaria parasites. Due to the high degree of DNA identity and amino acid similarity/identity, these genes likely arose due to a duplication event. It is notable that the same domain organization of 2 NOT1 paralogues is also present in the closely related apicomplexans *Babesia* and *Theileria*, while this duplication is not evident in *Toxoplasma* or other apicomplexans that have evolved other transmission strategies (Fig 1). To our knowledge, this duplication of NOT1 is unique to the Aconoidisida class. This duplication of NOT1 has been noted in preliminary findings in *P. falciparum* asexual blood stage parasites presented in a

**Table 1. Deletion of the TTP-binding domain of PyNOT1-G is important but dispensable for parasite transmission.**

| | WT-GFP (Py0489) | ΔTTPbd Clone 1 (Py1299) | ΔTTPbd Clone 2 (Py1299) |
|---|---|---|---|
| Prevalence of Infection | 91.7%/77.4%/59.5% | 2.9%/8.8%/24.7% | 3.2%/31.8%/42.6% |
| Average Oocysts/Infected Mosquito | 15.9/42.8/10.7 | 3.0/7.0/11.9 | 5.0/39.5/21.8 |

TTP, tristetraprolin.

recent preprint [43]. In that work, the transgenic *P. falciparum* parasites (and presumably the parental parasites used to generate them) could not generate gametocytes, and, thus, those studies were focused on discerning differences in their functions within the asexual blood stage. Here, we also observe effects upon asexual blood stage growth, which were extreme when *pynot1* is deleted but which were more muted when *pynot1-g* is deleted. In contrast, here, we show that the major essential roles of PyNOT1-G are in the development of male and female gametocytes/gametes and the continuing development of the fertilized zygote in early mosquito stage.

Consistent with their domain architecture and protein interactions, we found that PyNOT1 and PyNOT1-G localize to cytosolic puncta reminiscent of regulatory mRNP granules that are observed in many eukaryotes including *Plasmodium*. We and others have also observed this localization for CAF1, CCR4-1, ALBA4, DOZI, and CITH, which also interact with one another in specific contexts and life cycle stages [14,19,21]. Through extensive reverse genetic attempts, we were only able to isolate a completely transgenic parasite population where *py17x_1027900* (*pynot1*) was deleted, as the presence of any wild-type parasites would likely outcompete it rapidly. We now propose the reannotation of this gene as the canonical NOT1 as it matches the hallmarks of NOT1 from other eukaryotes, including an extremely slow growth phenotype when its gene is deleted [22]. It is interesting that *pynot1* could be deleted at all given its central and essential role in the activities of the CAF1/CCR4/NOT complex in model eukaryotes. This suggests that the organizing functions provided by the PyNOT1 scaffold can presumably be achieved by subcomplexes, or by proteins that can act in a functionally redundant manner, perhaps by PyNOT1-G, although this remains to be experimentally determined.

Here, we have focused on the NOT1 paralogue that does not behave like any other eukaryotic NOT1, which we have termed PyNOT1-G due to its essential roles in *Plasmodium* gametocytes and gametes. The timing of a slight asexual blood stage growth defect in *pynot1-g*⁻ correlates with the typical onset of a wave of commitment to gametocytogenesis and thus may correlate with the 27% to 35% increase in gametocyte numbers observed (Fig 3B). Furthermore, changes in transcript abundance in asexual blood stage schizonts depleted of gametocytes indicated that several mRNAs relevant to early gametocytogenesis are preserved by PyNOT1-G quite early on, and include transcripts for *male development gene 1* (*mdv1*), *actin ii*, *p25*, *p28*, *gest*, and several others. The ApiAP2-G specific transcription factor, which acts as a master regulator of commitment to gametocytogenesis was also dysregulated at the transcript level. However, we observed that *pyapiap2-g* mRNA levels were lower in both schizonts and gametocytes, which is discordant with the expectation that fewer gametocytes would form. Instead, we observed that gametocyte levels were slightly elevated (Figs 3B and 5). This would indicate that *pyapiap2-g* mRNA levels alone are not responsible for how extensively parasites commit to sexual development and that other factors contribute to this as well. Finally, we have intentionally used the general descriptor "preserve" when describing PyNOT1-G's effect upon transcript abundance, as it remains to be demonstrated for any *Plasmodium* protein implicated in translational repression or related complexes that a stabilizing effect is produced, although we hypothesize that this would be the case.

Transcriptomics of gametocytes also reflect that males are severely affected and is consistent with their inability to complete gametogenesis to produce exflagellating gametes. Many of the transcripts that are dysregulated in schizonts are also dysregulated in gametocytes and in some cases to an even greater extent. In considering female-enriched transcripts, there is a high overlap in the specific mRNAs that are dysregulated by a deletion of *pynot1-g*, *pyalba4*, *pbdozi*, or *pbcith* (S2 Table), and a common phenotype in female parasites when *pynot1-g*, *pbdozi*, or *pbcith* is deleted. This indicates that this level of female-enriched transcript dysregulation is

permissive to allow gametogenesis and fertilization by a competent male, but it appears to prevent further development of the zygote. This aligns with the classic model of the maternal-to-zygotic transition, where premade mRNAs are produced, stored, and translationally repressed/silenced until fertilization occurs [44].

Here, we find that PyNOT1-G primarily acts to promote male gametocytogenesis and/or gametogenesis as evidenced by the complete absence of male gametes and a transcriptomic profile that reflects severe dysregulation of male-enriched mRNAs (Fig 5). Unsurprisingly, male *pynot1-g⁻* parasites are also unable to transmit to mosquitoes (Fig 4A). Moreover, the inability of female parasites to produce viable zygotes that can develop into ookinetes in vitro (Fig 4B) or oocysts in vivo (Fig 4A) matches the phenotypes previously observed for deletions of *pbdozi* and *pbcith*. Together, it is clear that PyNOT1-G acts upon both sexes and matches or exceeds the phenotypes observed previously for the deletion of genes for its key interaction partners.

Our previous work demonstrated that both PyNOT1 and PyNOT1-G are capable of binding with the PyCCR4-1 deadenylase [21]. As NOT1 acts as a scaffold onto which other factors assemble, it is anticipated that NOT1 and NOT1-G are mutually exclusive members of discrete complexes, although this remains to be formally and robustly demonstrated. What is clear is that canonical CAF1/CCR4/NOT proteins, as well as possible ELAV/HuR and TTP candidates, all associate with this complex. As the functions of NOT1 in other eukaryotes can be toggled based upon the antagonistic association of ELAVs/HuRs or TTPs, our finding that this predicted TTP-binding domain is not important for NOT1-G functions in gametocyte biology is possibly surprising. One explanation for why *Plasmodium* may have evolved away from the use of this toggle in this rare NOT1 paralogue could be due to its extreme AT-rich genome and resulting AU-rich transcriptome. As ELAV/HuR and TTP typically bind with AU-rich elements on target mRNAs, this strategy would not provide much regulatory specificity in *Plasmodium*. Thus, different or additional parameters may be employed to ensure that on-target regulation occurs. Consistent with this hypothesis, we identified other zinc finger proteins associated with this complex, including one (PY17X_0417500) with a triple C3H1 ZnF configuration with predicted roles in mRNA decay. It will therefore be interesting to determine if and how these effector proteins associate with PyNOT1-G independently of the TTP-binding domain and to determine how these proteins can regulate only selected mRNAs given how frequently AU-rich mRNA sequences can be found in *Plasmodium*.

In order to synthesize our current understanding of how *Plasmodium* blood stage proteins implicated in translational repression (or with complexes known to be translationally repressive) may interact, we have reanalyzed published datasets to propose a composite network based upon formaldehyde crosslinking IP/MS datasets for PbDOZI::GFP, PbCITH::GFP, and PyALBA4::GFP from gametocytes and PyCCR4-1::GFP from schizonts [14,19,21] (Fig 7 and S5 Table). We also attempted to provide reciprocal proteomics data using PyNOT1-G as the bait (e.g., immunoprecipitation +/− chemical crosslinking, BioID, TurboID), but because these experiments did not provide additional insights using only highly stringent statistical cutoffs, we have not included them in this model. Moreover, it is notable that the phenotypes for *pynot1-g⁻* transgenic parasites are identical to those for *pbdozi⁻*/*pbcith⁻* (female gametocytes can activate into gametes and be fertilized but cannot develop further) and more extreme than those for *pyccr4-1⁻* or the truncation of *pycaf1* (complete arrest versus partial defects in either male gametocyte development or gametogenesis). Taken together with proteomic evidence, and consistent with the general role of NOT1 in model eukaryotes and humans, we propose that PyNOT1-G has been evolved to act as a central organizing nexus for these regulatory activities for both sexes of parasites, whereas DOZI/CITH and CCR4-1/CAF1 have sex-specific roles. Additionally, preliminary results from a preprint indicate that the *P. falciparum* NOT1

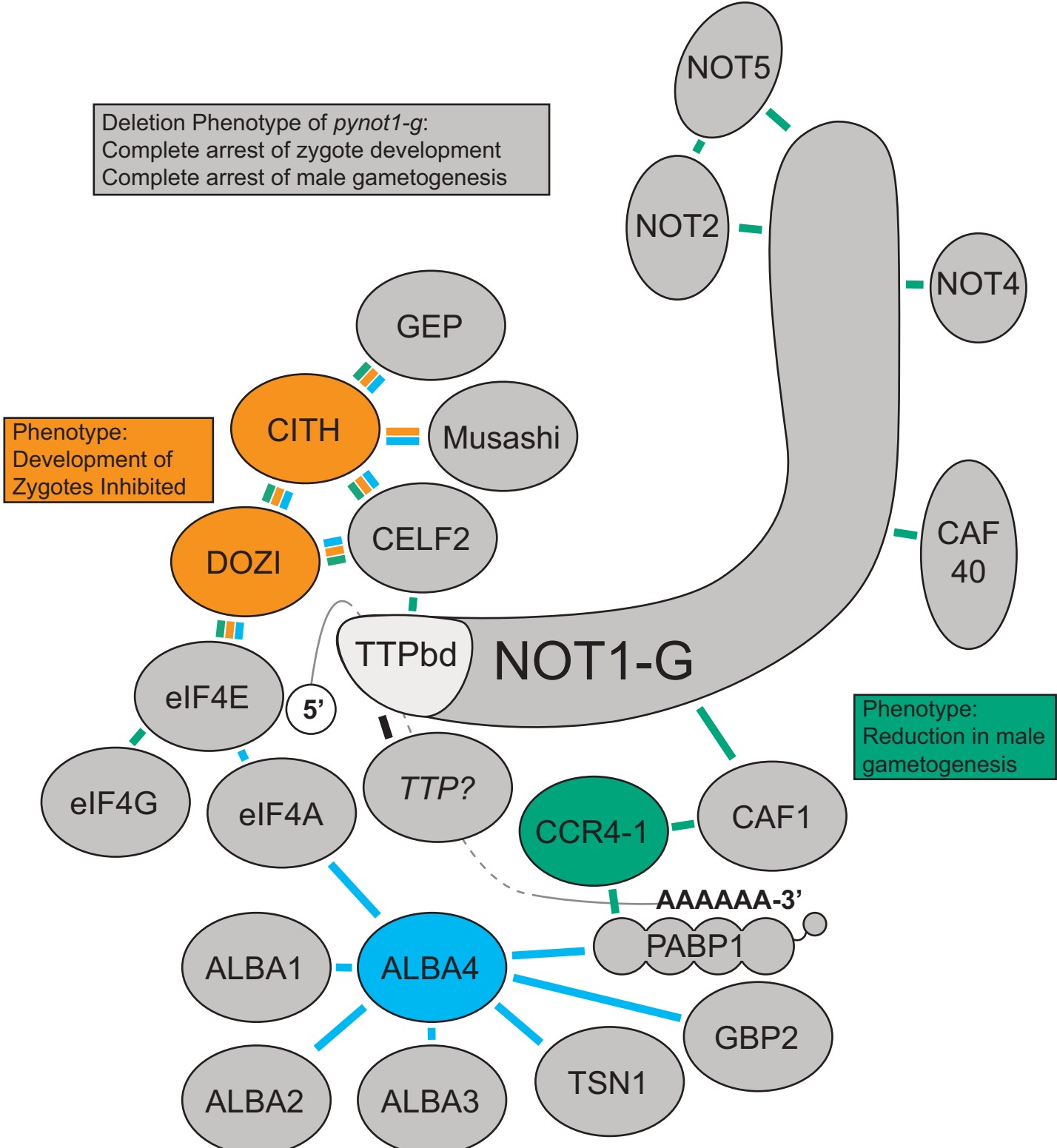

**Fig 7. A proposed composite interaction network of NOT1-G.** Crosslinking IP datasets for PbDOZI and PbCITH (orange lines), PyCCR4-1 (green lines), and PyALBA4 (light blue lines) were used to propose a composite interaction network with PyNOT1-G. The placement of some proteins is based upon known direct interactions from the literature (e.g., NOT1-G, CAF1, CCR4, CAF40, NOT4, NOT2, NOT5, PABP1, CELF2, DOZI, eIF4F (eIF4E, eIF4A, eIF4G)), as are the placements of the 5′ m7G cap next to the cap-binding protein eIF4E and the poly(A) tail with PABP1. Deletion phenotypes for *pynot1-g* match or exceed those of *pbdozi⁻*, *pyccr4–1⁻*,

and the *pycaf1* truncation and lead to a complete arrest in male parasite development and in development of female parasites beyond the zygote stage. TTP, tristetraprolin.

and NOT1-G orthologues are mutually exclusive in their respective complexes [43]. This would be expected, as the classic role of NOT1 in eukaryotes is to act as a scaffold upon which effector proteins can be recruited. Moreover, if this hypothesis holds true, another possible mechanism by which PyNOT1-G could function would be to act as a decoy by sufficiently sponging effector proteins away from PyNOT1. However, because the localization and approximate expression levels of PyNOT1 and PyNOT1-G are effectively equal, a stronger interaction between PyNOT1-G and these proteins would be required. Because crosslinking was used in proteomic studies of proteins known to associate with this complex, this permits the detection of direct, short-distance, and long-distance interactions. However, future studies with proximity proteomics approaches could define the spatial and temporal parameters of this complex. Hints as to how these granules may be organized and regulated in *Plasmodium* are perhaps being found in work with yeast. First, a recent study indicated that members of the CCR4/NOT complex are present in processing bodies but were found at lower concentrations and were more dynamic [45]. Additionally, 2 other studies indicated that the partitioning of Dhh1, the ortholog of DOZI, is adversely affected by the association with NOT1 and positively affected by association with Pat1 [30,46]. Because no bioinformatically predictable ortholog of Pat1 is present in any *Plasmodium* species, identifying and understanding the regulators of granule assembly and disassembly will require further experimentation.

Overall, we conclude that *Plasmodium* and closely related parasites have duplicated NOT1 to allow for the dedication of one paralogue for a gametocyte/gamete-specific role in host-to-vector transmission. However, key aspects of how PyNOT1-G toggles between its dual natures remain unanswered. As there is an overrepresentation of AU-rich sequences in the *Plasmodium* transcriptome, the role of ELAV/HuR and TTP RNA-binding proteins that typically bind these AU-rich sequences may be diminished. Accordingly, the TTP-binding domain that typically enables their recruitment to the CAF1/CCR4/NOT complex in model eukaryotes and human cells is dispensable for PyNOT1-G's critical functions. Therefore, it is plausible that *Plasmodium* has adopted an alternate strategy by adapting the functions of canonical binding partners or through the recruitment of additional *Plasmodium*-specific proteins to this unique, transmission-adapted NOT1 paralogue.

## Materials and methods

### Ethics statement

All animal care strictly followed the Association for Assessment and Accreditation of Laboratory Animal Care (AAALAC) guidelines and was approved by the Pennsylvania State University Institutional Animal Care and Use Committee (IACUC# PRAMS201342678). All procedures involving vertebrate animals were conducted in strict accordance with the recommendations in the Guide for Care and Use of Laboratory Animals of the National Institutes of Health with approved Office for Laboratory Animal Welfare (OLAW) assurance.

### Experimental animals

Six-to-eight-week-old female Swiss Webster (SW) mice from Envigo were used for all experiments in this work. *Anopheles stephensi* mosquitoes were reared at 24 C and 70% humidity and were fed with 0.05% w/v PABA-supplemented 10% w/v sugar water.

## Production and confirmation of *Plasmodium yoelii* transgenic parasites

Transgenic *P. yoelii* 17XNL strain parasites were produced through conventional reverse genetics approaches using 2 homology arms that are specific to the targeted gene [47,48]. Homology arms were PCR amplified from wild-type genomic DNA, were combined into a single PCR amplicon by SOE PCR, were inserted into pCR-Blunt for sequencing, and were finally inserted into a pDEF plasmid for use in *P. yoelii* parasites. Plasmids used for gene deletion were based upon pSL0444, which replaces the gene sequences with a GFPmut2 expression cassette, a HsDHFR expression cassette, and the plasmid backbone. Plasmids used for appending a C-terminal GFPmut2 tag to a protein were based upon pSL0442, which inserts a the coding sequence for GFPmut2 with the *P. berghei* DHFR 3′ UTR, a HsDHFR expression cassette, and the plasmid backbone into that genomic locus. The PyWT-GFP transgenic line (Py0489) has been described previously [14], and the PyWT-mScarlet line (Py1115) was created in an identical manner but substitutes mScarlet coding sequences for those of GFPmut2. Overexpression of the TTP-binding domain of PyNOT1-G was achieved by inserting these coding sequences (AA1-199) upstream of the GFPmut2 sequences of Py0489. The deletion of the TTP-binding domain of PyNOT1-G was achieved by substitution of these coding sequences with those of GFPmut2 in an N-terminal tag format. Primers used to create all amplicons are provided in S6 Table, and complete plasmid sequences used to create PyWT-GFP, PyWT-mScarlet, *pynot1⁻*, *pynot1-g⁻*, *pynot1::gfp*, *pynot1-g::gfp*, overexpressor of the TTP-binding domain, and the deletion of the TTP-binding domain are provided in S1 File.

Schizonts used for transfections were produced via ex vivo cultures and were purified by an Accudenz discontinuous gradient as previously described [49]. Purified parasites enriched in schizonts were transfected with 10 μg linearized plasmid using an Amaxa Nucleofector 2b device using either Lonza T-Cell Solution with Program U-033 or with cytomix using Program T-016. Transgenic parasites were selected by pyrimethamine drug cycling in a parental and a transfer mouse. In some cases, parasites were enriched by FACS and/or cloned by limited dilution approaches. The genotype of parasites was determined by PCR across both homology arms for the targeted genomic locus.

## Production and Accudenz purification of *P. yoelii* schizonts and gametocytes

Schizonts that were depleted of rings, trophozoites, and gametocytes were produced through a subtractive magnetic approach. Blood was collected from mice at approximately 1% to 2% parasitemia into RPMI1640 media and was passed over a magnetic column (Miltenyi Biotec, LS Columns) to capture late trophozoites, schizonts, and gametocytes by virtue of the magnetic hemozoin present in them. In contrast, rings and early trophozoites flowed through the column and were collected into fresh media. This depletion process was repeated a total of 4 times, after which no further depletion of gametocytes in the flow through was observed. Parasites were then subjected to ex vivo culture for 12 to 14 hours until mature schizonts were visible by Giemsa-stained thin blood smears.

Gametocytes were produced by treatment of the mice at 1% parasitemia with 10 mg/L sulfadiazine (VWR, Cat# AAA12370-30) in their drinking water for 2 days before exsanguination. The blood was placed in 30 mL of prewarmed RPMI1640 with 20% v/v FBS to prevent activation of gametocytes and was purified using an Accudenz gradient as previously described [14].

## Live fluorescence and IFA-based microscopy

PyNOT1 and PyNOT1-G expression in blood stages, oocyst sporozoites, salivary gland sporozoites, and liver stages was investigated by an indirect immunofluorescence assay (IFA), and expression in day 7 oocysts was observed by live fluorescence microscopy. All samples for IFA

were prepared as previously described [47]. Parasites were stained with the following primary antibodies: rabbit anti-GFP (1:1,000, Invitrogen, Cat# A11122; 1:1,000, Pocono Rabbit Farm & Laboratory, Custom polyclonal antibody), rabbit anti-PyACP (1:1,000, Pocono Rabbit Farm & Laboratory, Custom polyclonal antibody), mouse anti-GFP (1:1,000, DSHB, Clone 4C9), rabbit anti-HsDDX6 that cross-reacts with DOZI (1:1,000, gift from Joe Reese, Custom PAb), mouse anti-alpha-tubulin (Clone B-5-1-2) (1:1,000, Sigma, Cat# T5168), and mouse anti-PyCSP (1:1,000, Clone 2F6 [50]). Secondary antibodies used for all stages were Alexa Fluor conjugated (AF488, AF594) and specific to rabbit or mouse (1:1,000, Invitrogen, Cat# A11001, A11005, A11008, A11012). 4′,6-diamidino-2-phenylindole (DAPI) was used to stain nucleic acids following washing away unbound secondary antibodies, and samples were covered with VectaShield anti-fade reagent (Vector Laboratories, VWR, Cat# 101098–048) and a coverslip, then sealed with nail polish before visualization. Fluorescence and DIC images were taken using a Zeiss fluorescence/phase contrast microscope (Zeiss Axioscope A1 with 8-bit AxioCam ICc1 camera) using a 40X or 100X oil objective and processed by Zen imaging software.

## RNA-seq sample production, data analysis, and interpretation

RNA from *P. yoelii* asexual blood stage schizonts or gametocytes was prepared using the Qiagen RNeasy kit with 2 sequential DNaseI on-column digests, with quality control analysis by BioAnalyzer. Samples were used to create barcoded libraries (Illumina TruSeq Stranded mRNA Library) and were sequenced on an Illumina HiSeq 2500 to yield 100 nt long single end reads for each of 3 biological replicates per sample type. These data were mapped to the *P. yoelii* reference genome (*P. yoelii* 17X strain, plasmodb.org v50) using hisat2 (version 2.1.0) [51] specifying—rna-strandness R and—max-intronlen 5000 parameters. Coverage files were generated and the mapped data were visualized and manually inspected in Integrative Genomics Viewer (IGV) as a quality control check [52]. Reads that mapped to annotated genes (PlasmoDB.org v50) were counted using featureCounts (version 2.0.0) [53] and specified–s 2 –t exon–g gene_id parameters. The raw read counts were normalized with DESeq2 (version 1.26.0) [54], which was also used to determine the differentially expressed genes between the sample types. The TIN [40] for each gene was calculated and used to further filter out the differentially expressed genes that do not have coverage evenness across the entire length of the gene as described before [39]. Briefly, TIN was calculated from the transcriptome as built from the annotation files and pseudobam files as generated using Kallisto (version 0.46.1, parameters used: specifying—rf-stranded—single -l 150 -s 20 -b 10) [55]. The samtools depth command was used to determine read depth at each position in the transcript, and TIN was calculated as previously described [55]. Custom python scripts are provided as S3 File. For genes with multiple transcripts, the maximum of the transcript TINs was considered as the integrity value for the gene. Differentially expressed genes with FDR $<0.05$ were flagged for inspection if the read count is $>20$ in either of the conditions, but TIN is $<40$ in both conditions. TIN-$\log_2$-fold change was calculated, and the flagged genes were resolved and included in the differential expression set if the absolute value of TIN $\log_2$-FC is $< = 1.5$. For genes with read count $>20$ and TIN $<40$, transcript overlap from neighboring genes were checked using a custom python script (S3 File) and were flagged if an overlap was detected. A MakeFile describing the complete bioinformatics workflow (including all steps noted above) and other scripts used are also provided in S3 File. GO terms (cellular components, molecular functions, and biological processes) were retrieved from PlasmoDB.org (v51). GO terms for the differentially expressed transcripts were summarized using REVIGO and were plotted using CirGO, with GO terms that are single clusters labeled as null [56,57]. RNA-seq data reported here are available through the GEO depository (Accession #GSE136674). To generate a heatmap of 50

representative transcripts affected by the deletion of *pynot1-g*, DESeq2 normalized counts were converted to a z-score to denote value deviation from the mean. Positive values (red) and negative values (green) denote an increase or decrease in abundance in *pynot1-g*⁻ parasites, respectively. A PCA plot was generated with the plotPCA function available in DESeq2 package, using VST with transformed raw counts as the input. To generate MA plots, shrunken log$_2$ fold changes were estimated using adaptive shrinkage estimator from the ashr package [58]. These were then visualized with plotMA function in DESeq2 package. Points are colored red if the adjusted *p*-value is less than 0.05. Data points that fall out of these limits are plotted as open triangles pointing either up or down.

## Measurement of asexual growth kinetics and male gametocyte activation

Cryopreserved blood infected with either *P. yoelii* wild-type (Py17XNL strain) or transgenic parasites was injected intraperitoneally into SW starter mice, and parasitemia was allowed to increase to 1%. This blood was extracted via cardiac puncture and diluted in RPMI to 1,000 parasites per 100 μl. Intravenously (IV), 100 μl was injected into 3 mice for each of 3 biological replicates for each parasite line. Parasitemia was measured daily by Giemsa-stained thin blood smears, and male gametocyte activation was monitored daily by counting exflagellation centers ("centers of movement") via wet mount of a drop of blood incubated at room temperature for 8 minutes as previously described (21).

## Measurement of differences in gametocytemia by flow cytometry

Comparisons of gametocyte numbers were made using flow cytometry. Briefly, 3 experimental mice were infected with 1,000 PyWT-GFP or *pynot1-g*⁻ (GFP+) parasites by IV injection as described above. The parasitemia was monitored daily, and upon reaching 1% parasitemia, the mice were treated with sulfadiazine for 2 days to selectively kill asexual blood stage parasites. Blood was collected by cardiac puncture, and the fraction of GFP+ cells were counted on a Beckman Coulter Astrios Moflo EQ (Brea, CA) in tube mode using uninfected blood and *P. yoelii* 17XNL wild-type parasite-infected blood as negative controls. All data were analyzed by FlowJo (v10.6.1).

## Mosquito transmission

SW mice were infected with *P. yoelii* wild-type or transgenic parasites by IP or IV injection as previously described [21]. Mice were screened daily for parasitemia and the presence of male gametocyte activation (centers of movement). On the time points indicated, the mice were anesthetized by IP injection of a ketamine/xylazine cocktail and were exposed to mosquitoes for 15 minutes with their positions adjusted every 5 minutes to allow for more even feeding. Mosquito transmission was assessed by dissection of a minimum of 50 midguts on day 7 post-blood meal, which were analyzed for the prevalence of infection and oocyst numbers by DIC and fluorescence microscopy.

## Genetic crosses

**WT (colorless) versus *pynot1-g*⁻ (GFP+) genetic cross.**   Blood from mice infected with either Py17XNL wild-type parasites or *pynot1-g*⁻ transgenic parasites was serially diluted to 5,000 infected red blood cells per 50 μl and was used to infect groups of 3 experimental mice with either 10,000 wild-type parasites, 5,000 wild-type parasites and 5,000 *pynot1-g*⁻ transgenic parasites, or 10,000 *pynot1-g*⁻ transgenic parasites in biological triplicate. Parasitemia and male gametocyte activation were monitored daily, and mice were fed to mosquitoes as above

on the day indicated. Midguts were analyzed on day 7 post-blood meal on a fluorescence microscope as above, and numbers of oocysts and their fluorescence were recorded (green or colorless).

**PyWT-mScarlet, PyWT-GFP, *pynot1-g⁻* (GFP+).** Blood from mice infected with either Py17XNL wild-type parasites expressing mScarlet (PyWT-mScarlet, Py1115) or GFPmut2 (PyWT-GFP, Py489) or *pynot1-g⁻* transgenic parasites was collected by cardiac puncture and combined for in vitro ookinete cultures as previously described [59,60]. Briefly, blood was collected on the peak day of exflagellation for Py489 and/or Py1115 (as male *pynot1-g⁻* transgenic parasites do not complete gametogenesis and do not exflagellate) when the parasitemia was each of the mice was comparable. Blood from each infected mouse was added to 30 ml iRPMI (RPMI with HEPES and L- glutamine; no FBS), and infected red blood cells were enriched by an Accudenz gradient as described above. The parasitized cells were collected from the interface and resuspended in iRPMI. For genetic cross experiments, equal volumes of resuspended parasites at comparable parasitemia were either added alone or in combination with another parasite line to 10 ml in vitro ookinete media (RPMI with HEPES and L-glutamine, 20% FBS (Corning, Prod No. 35-011-CV, Lot No. 35011126), 0.05% w/v hypoxanthine, 100 μM xanthurenic acid (pH 8.2)) and were allowed to develop at room temperature for 24 hours. Parasites were assessed by live fluorescence microscopy for mScarlet and/or GFPmut2 expression as described above.

## Statistical analyses

Statistical tests for RNA-seq analyses and comparisons were provided by DEseq2 [54]. The statistical test assessments of GO terms (odds ratio, *p*-value, Benjamini, Bonferroni) were provided by PlasmoDB.org (v51). All other statistical tests were conducted with GraphPad Prism (v9) with the specific test(s) used noted where applied.

## Supporting information

**S1 Fig. Genotyping PCR of PyNOT1::GFP and PyNOT1-G::GFP transgenic parasites.** Genomic DNA from transgenic parasites was compared to that of Py17XNL wild-type parasites or no template controls by genotyping PCR. A schematic of the wild-type and designed transgenic loci are provided below each gel image. Primers used are identified in S6 Table for assessing (A) PyNOT1::GFP and (B) PyNOT1-G::GFP parasites. (A) The PSU 1 kb MW ladder [61] or the NEB 1 kb+ MW ladder (B) flanks all experimental lanes. MW, molecular weight. (PDF)

**S2 Fig. PyNOT1 and PyNOT1-G have similar expression patterns throughout mosquito stage development.** The expression of PyNOT1::GFP and PyNOT1-G::GFP in mosquito stage were assessed by live fluorescence (day 7 oocysts) or by IFA (day 10 oocyst sporozoites, day 14 salivary gland sporozoites). Sporozoites were counterstained with anti-PyCSP and DAPI. Scale bar is 20 μm (oocyst) or 10 μm (sporozoites). DIC, differential interference contrast; IFA, immunofluorescence assay. (PDF)

**S3 Fig. Genotyping PCR of *pynot1⁻* and *pynot1-g⁻* transgenic parasites.** Genomic DNA from (A) *pynot1⁻* or (B) *pynot1-g⁻* transgenic parasites was compared to that of Py17XNL wild-type parasites or no template controls by genotyping PCR. A schematic of the wild-type and designed transgenic loci are provided below each gel image. Primers used are identified in S6 Table. (A) The NEB 1 kb+ MW ladder (B) flanks all experimental lanes. MW, molecular weight. (PDF)

**S4 Fig. Genotyping PCR and western blotting of TTPbd::GFP overexpression parasites and ΔTTPbd transgenic parasites. (A, C)** Genomic DNA from transgenic parasites was compared to that of Py17XNL wild-type parasites or no template controls by genotyping PCR. A schematic of the wild-type and designed transgenic loci are provided below each gel image. (B) Western blotting of TTPbd::GFP vs Py17XNL WT-GFP parasite lysate enriched by immunoprecipitation with anti-GFP is shown (I = Input, FT = Flow Through, E = Elution). Primers used are identified in S6 Table for assessing (A) PyNOT1-G TTPbd::GFP at the *p230p* genomic locus and (B) PyNOT1-G::GFP parasites. (A) The PSU 1 kb MW ladder [61] or the NEB 1 kb + MW ladder (C) flanks all experimental lanes. MW, molecular weight; TTP, tristetraprolin. (PDF)

**S1 Table. Processed RNA-seq data for gametocyte samples comparing Py17XNL wild-type parasites and *pynot1-g⁻* transgenic parasites as conducted in biological triplicate.** (XLSX)

**S2 Table. Overlapping genes that are similarly dysregulated in *pynot1-g⁻*, *pyalba4⁻*, *pbdozi⁻*, and *pbcith⁻* transgenic parasites.** (XLSX)

**S3 Table. Processed RNA-seq data for asexual blood stage schizont samples comparing Py17XNL wild-type parasites and *pynot1-g⁻* transgenic parasites as conducted in biological triplicate.** (XLSX)

**S4 Table. Transmission data for Py17XNL WT-GFP parasites (Py489) and PyNOT1-G TTPbd::GFP overexpression parasites (Py1125).** (XLSX)

**S5 Table. Comparisons of immunoprecipitation/mass spectrometry–based proteomic datasets for PyCCR4-1, PyALBA4, and PbDOZI/PbCITH (related to Fig 7).** (XLSX)

**S6 Table. Oligonucleotides used in this study.** (XLSX)

**S1 File. Complete sequences for plasmids used in this study.** (DOCX)

**S2 File. Flow cytometry panels obtained during the measurement of gametocytemia of PyWT-GFP and *pynot1-g⁻* transgenic parasites (related to Fig 3B).** (PDF)

**S3 File. A Makefile log of complete RNA-seq bioinformatics workflow, along with all custom R and python scripts used in this work.** (ZIP)

**S1 Data. The raw data for Figs 3A, 3B, 3C, 4A, 6A, 6B, 6C, and 6D, and S4 Table are provided.** (XLSX)

**S1 Raw Images. The raw gel image files for gel-based data presented in this publication are provided.** (TIF)

## Acknowledgments

We acknowledge Andy Waters for helpful discussions in the depletion of gametocytes from mixed blood stage parasites, and Katarzyna Modrzynska for advice on the optimization of in vitro ookinete cultures. We are grateful to Sarah Neering and Brian Dawson for assistance with flow cytometry data collection and analysis, as well as to the Penn State Genomics and Flow Cytometry Facilities operated by the Huck Institutes of the Life Sciences, University Park, Pennsylvania. We acknowledge and appreciate the critical guidance from Istvan Albert on the processing and analysis of RNA-seq datasets. Finally, we thank members of the Lindner and Llinás laboratories for critical discussions and evaluations of this manuscript.

## Author Contributions

**Conceptualization:** Kevin J. Hart, Kelly T. Rios, Scott E. Lindner.

**Data curation:** Kevin J. Hart, Scott E. Lindner.

**Formal analysis:** Kevin J. Hart, Kelly T. Rios, Aswathy Sebastian, Scott E. Lindner.

**Funding acquisition:** Scott E. Lindner.

**Investigation:** Kevin J. Hart, B. Joanne Power, Kelly T. Rios, Scott E. Lindner.

**Methodology:** Kevin J. Hart, B. Joanne Power, Kelly T. Rios, Scott E. Lindner.

**Project administration:** Scott E. Lindner.

**Software:** Aswathy Sebastian.

**Supervision:** Scott E. Lindner.

**Validation:** Scott E. Lindner.

**Visualization:** Aswathy Sebastian, Scott E. Lindner.

**Writing – original draft:** Kevin J. Hart, Kelly T. Rios, Aswathy Sebastian, Scott E. Lindner.

**Writing – review & editing:** Kevin J. Hart, Kelly T. Rios, Aswathy Sebastian, Scott E. Lindner.

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
