## [Editor Report · Decision Letter 0]

16 Aug 2021

Dear Dr. Lindner, 

Thank you for submitting your manuscript entitled "The Plasmodium NOT1-G Paralogue Acts as an Essential Nexus for Sexual Stage Maturation and Parasite Transmission" for consideration as a Research Article by PLOS Biology.

Your manuscript has now been evaluated by the PLOS Biology editorial staff, as well as by an academic editor with relevant expertise, and I am writing to let you know that we would like to send your submission back to the original reviewers.

Please re-submit your manuscript within two working days, i.e. by Aug 18 2021 11:59PM.

Kind regards,

Paula

---

Paula Jauregui, PhD

Associate Editor

PLOS Biology

---

## [Decision Letter · Decision Letter 1]

3 Sep 2021

Dear Dr. Lindner,

Thank you very much for submitting a revised version of your manuscript "The Plasmodium NOT1-G Paralogue Acts as an Essential Nexus for Sexual Stage Maturation and Parasite Transmission" for consideration as a Research Article at PLOS Biology. This revised version of your manuscript has been evaluated by the PLOS Biology editors, the Academic Editor and the original reviewers.

The reviewers, mainly reviewer #3, still have some issues that need to be addressed. You would need to do the bioinformatic analyses requested by the reviewer, and prove that your alpha tubulin staining pattern is unique to male gametocytes. The Academic Editor has The Academic Editor has kindly provided some guidance as to how you should address the remaining comments from reviewer #3. You can find those comments at the end of this letter. 

In light of the reviews (below), we will not be able to accept the current version of the manuscript, but we would welcome re-submission of a much-revised version that takes into account the reviewers' comments. We cannot make any decision about publication until we have seen the revised manuscript and your response to the reviewers' comments. Your revised manuscript is also likely to be sent for further evaluation by the reviewers.

We expect to receive your revised manuscript within 3 months. 

**IMPORTANT - SUBMITTING YOUR REVISION**

*Re-submission Checklist*

*Published Peer Review*

*PLOS Data Policy*

*Blot and Gel Data Policy*

Sincerely,

Paula

---

Paula Jauregui, PhD

Associate Editor

PLOS Biology

REVIEWS:

Reviewer #2: Sebastian Baumgarten. Plasmodium RNA biology.

Reviewer #3: Molecular parasitology.

Reviewer #2: The authors have addressed all my previous comments and I am very much looking forward to seeing this work published.

Two minor comments: Please add the datapoints to the bar graph in Figure 4A and make sure that the quality of the figures is ok (which could also be the manuscript review system)

Reviewer #3: This new version has been significantly improved compared with the previous one.

I however think that:

- including discussions in the result section is a way to over interpret the results;

- the visual representation of the transcriptomic results is mainly showing the number of regulated genes but not highlighting the biology behind the dots;

- the authors over-interpret the localization of Not1-G and do not reach the limit of fluorescence microscopy, as stated in their response. At this resolution and in the absence of co-localisation, many proteins unrelated to the CAF1/CCR4/NOT complex will show a similar localization pattern. Importantly, the authors would have to demonstrate that the apparent nuclear localization of alpha-tubulin in male gametocytes is linked to non-confocal imaging. This is, at least, not the case in our hands.

COMMENTS FROM THE ACCADEMIC EDITOR:

The reviewer is not satisfied with the visual presentation of the rnaseq data. GO analysis is covered in the text and the full dataset is available in the supplementary tables but the reviewer requested a figure for the GO, something like CirGO would be very easy to make and would reduce complexity and clarify how specific to the described genes and genesets the dysregulation is, ie whether there were lots of other GO categories also dysregulated. I am a little surprised at the authors' response because the GO analysis seems to nicely support their hypotheses. I support the reviewer here as readers should be able to assess how valid the conclusions re GO enrichment were without consulting supplementary tables. I don’t think that including a simple CirGO or similar plot would overwhelm fig 5 which is currently carrying 2 essentially redundant panels that use orthologous approaches to demonstrate reproducibility of the replicates. These are the PCA plot and correlation heatmap of the replicates. The reviewer initially asked for heatmaps of expression of the genes discussed in the text, not the heatmap of the correlation between replicates provided by the authors. Although it is nice to include evidence of the reproducibility of the rnaseq. I would suggest moving the pca or the correlation heatmap to the supplement and adding the heatmap of gene expression that was requested by the reviewer as well as a visual representation of the disregulated GOs.

The RNASeq data in Fig 5 is also lacking a clear labelling of the bland altman/MA plots in fig 5 c and d, these are not labelled adequately nor explained in the legend, log fold change of what? Presumably its KD/wildtype but this needs to be explained in the legend or figure. Similarly the volcano plots in 5 e and d are not explained, again I presume its KO/wildtype but this needs to be explained in the legend or the figure.

The RNAseq is described in lines 253 – 268 but the authors describe fold change but not p values or FDR, this is provided in the supplementary tables but they should explicitly state the threshold for significance for the fold changes they describe in the text.

The reviewer is unconvinced by the fluorescence microscopy of not-1. I share their reservations about the cytoplasmic pattern being indicative of an association with caf1 and ccr4 but I agree with the authors that the staining is partially punctate. I suggest they moderate their interpretation to state that the staining pattern of not-1 is cytoplasmic and partially punctate consistent with other cytoplasmic proteins including the previously observed patterns of ccr4 and caf1. As the authors themselves note the co-IP data is what shows an interaction between not1 and the ccr4 complex components, the imaging is not very informative and doesn’t need to be over-interpreted.

It’s unclear to me whether the reviewer is concerned that the alpha tubulin staining is non-specific and therefore the parasite may not be a male gametocyte, or whether they are concerned that the quality of the fluorescence microscopy is inadequate because of the depth of field captured. The former then this is a genuine concern that should be addressed by staining more cells with anti alpha tubulin and showing that only male gametocytes are stained, preferably also staining with an additional male gam specific antibody. Failing that presumably the authors have some negative control asexual parasites they stained at the same that they could show. If it is the latter concern then I think it is a technical issue that doesn’t really affect the interpretation of the data, ie the not 1 stain is clearly outside the nucleus so the parasite has not1 outside the nucleus.

---

## [Editor Report · Decision Letter 2]

22 Sep 2021

Dear Dr Lindner,

I am writing on behalf of my colleague Paula Jauregui, who is currently on annual leave. 

Thank you for submitting your revised Research Article entitled "The Plasmodium NOT1-G Paralogue Acts as an Essential Nexus for Sexual Stage Maturation and Parasite Transmission" for publication in PLOS Biology. We have discussed your new version and your response to reviewers with the Academic Editor, and I am pleased to tell you that we will probably accept this manuscript for publication, provided you satisfactorily address the following points:

1) Title:

We would like to recommend a title that would be more accessible to the broad readership of PLOS Biology:

"The Plasmodium NOT1-G paralogue is an essential regulator of sexual stage maturation and parasite transmission."

or

"The Plasmodium NOT1-G paralogue regulates RNAs that are essential for sexual stage maturation and parasite transmission"

2) Intentional design: 

You should modify throughout the manuscript all references to intentional design. For example, in the abstract you write, “Here we show that Plasmodium parasites have taken the unique approach to duplicate the NOT1 scaffold protein of the CAF1/CCR4/Not complex in order to dedicate one paralogue for essential transmission functions.” and “Together, we conclude that Plasmodium has created and adapted a NOT1-G paralogue to fulfill the complex transmission requirements of both male and female parasites.” We suggest the following, “Here we show that in Plasmodium parasites the NOT1 scaffold protein of the CAF1/CCR4/Not complex is duplicated and one paralogue is dedicated for essential transmission functions.” and “Together, we conclude that a NOT1-G paralogue in Plasmodium fulfills the complex transmission requirements of both male and female parasites.”

3) Data:

Note that we do not require all raw data. Rather, we ask for all individual quantitative observations that underlie the data summarized in the figures and results of your paper. For an example see here: http://www.plosbiology.org/article/info%3Adoi%2F10.1371%2Fjournal.pbio.1001908#s5

These data can be made available in one of the following forms:

i) Supplementary files (e.g., excel). Please ensure that all data files are uploaded as 'Supporting Information' and are invariably referred to (in the manuscript, figure legends, and the Description field when uploading your files) using the following format verbatim: S1 Data, S2 Data, etc. Multiple panels of a single or even several figures can be included as multiple sheets in one excel file that is saved using exactly the following convention: S1_Data.xlsx (using an underscore).

ii) Deposition in a publicly available repository. Please also provide the accession code or a reviewer link so that we may view your data before publication.

Regardless of the method selected, please ensure that you provide the individual numerical values that underlie the summary data displayed in the following figure panels: Figures 3A-C, 4AB, 5A-H, and 6A-D.

Please also ensure that each figure legend in your manuscript includes information on where the underlying data can be found and that your supplemental data file/s has/have a legend.

4) Gels:

We require the original, uncropped and minimally adjusted images supporting all blot and gel results reported in an article's figures or Supporting Information files. We will require these files before a manuscript can be accepted so please prepare and upload them now. Please carefully read our guidelines for how to prepare and upload this data: https://journals.plos.org/plosbiology/s/figures#loc-blot-and-gel-reporting-requirements

5) Data not shown:

Please note that per journal policy, we do not allow the mention of "data not shown", "personal communication", "manuscript in preparation" or other references to data that is not publicly available or contained within this manuscript. Please either remove mention of these data or provide figures presenting the results and the data underlying the figure(s).

We expect to receive your revised manuscript within two weeks. 

*Published Peer Review History*

*Early Version*

Sincerely,

Gabriel Gasque on behalf of

Editor,

pjaureguionieva@plos.org,

PLOS Biology

---

## [Editor Report · Decision Letter 3]

4 Oct 2021

Dear Dr. Lindner,

On behalf of my colleagues and the Academic Editor, Michael Duffy, I am pleased to say that we can in principle offer to publish your Research Article "The Plasmodium NOT1-G paralogue is an essential regulator of sexual stage maturation and parasite transmission" in PLOS Biology, provided you address any remaining formatting and reporting issues. These will be detailed in an email that will follow this letter and that you will usually receive within 2-3 business days, during which time no action is required from you. Please note that we will not be able to formally accept your manuscript and schedule it for publication until you have made the required changes.

PRESS

Sincerely, 

Paula 

---

Paula Jauregui, PhD 

Associate Editor 

PLOS Biology
